# Finding structure during incremental speech comprehension

**Bingjiang Lyu[1]\*, William D Marslen-Wilson[2], Yuxing Fang[2], Lorraine K Tyler[2]\***

[1]Changping Laboratory, Beijing, China; [2]Centre for Speech, Language and the Brain, Department of Psychology, University of Cambridge, Cambridge, United Kingdom

**Abstract** A core aspect of human speech comprehension is the ability to incrementally integrate consecutive words into a structured and coherent interpretation, aligning with the speaker's intended meaning. This rapid process is subject to multidimensional probabilistic constraints, including both linguistic knowledge and non-linguistic information within specific contexts, and it is their interpretative coherence that drives successful comprehension. To study the neural substrates of this process, we extract word-by-word measures of sentential structure from BERT, a deep language model, which effectively approximates the coherent outcomes of the dynamic interplay among various types of constraints. Using representational similarity analysis, we tested BERT parse depths and relevant corpus-based measures against the spatiotemporally resolved brain activity recorded by electro-/magnetoencephalography when participants were listening to the same sentences. Our results provide a detailed picture of the neurobiological processes involved in the incremental construction of structured interpretations. These findings show when and where coherent interpretations emerge through the evaluation and integration of multifaceted constraints in the brain, which engages bilateral brain regions extending beyond the classical fronto-temporal language system. Furthermore, this study provides empirical evidence supporting the use of artificial neural networks as computational models for revealing the neural dynamics underpinning complex cognitive processes in the brain.

**\*For correspondence:**
bingjiang.lyu@gmail.com (BL);
lkt10@cam.ac.uk (LKT)

**Competing interest:** The authors declare that no competing interests exist.

## eLife assessment

This **valuable** study provides insights into how the brain parses the syntactic structure of a spoken sentence. **Convincing** evidence is provided that distributive cortical networks are engaged for incremental parsing of a sentence, and neural activity recorded by MEG correlates with sentence structure measures extracted by a deep neural network language model, that is, BERT. A contribution of the work is to use a deep neural network model to quantify how the mental representation of syntactic structure updates as a sentence unfolds in time.

## Introduction

Human speech comprehension involves a complex set of processes that transform an auditory input into the speaker's intended meaning, wherein each word is sequentially recognized and integrated with the preceding words to obtain a coherent interpretation (*Marslen-Wilson and Tyler, 1980*; *Choi et al., 2021*; *Lyu et al., 2019*). Crucially, rather than simple linear concatenation, individual words are combined according to the non-linear and often discontinuous structure embedded in an utterance as it is delivered over time (*Everaert et al., 2015*). For example, in the sentence "*The boy who chased the cat was …*", it is the structurally close word '*boy*', rather than the linearly close word '*cat*', that is combined with '*was*'. However, the neural dynamics underpinning the incremental construction of a structured interpretation from a spoken sentence is still unclear.

Previous neuroimaging studies on the structure of language primarily focused on syntax (*Matchin and Hickok, 2020*), contrasting grammatical sentences against word lists or sentences with syntactic violations (*Law and Pylkkänen, 2021*; *Nelson et al., 2017*), manipulating the syntactic complexity in sentences (*Pallier et al., 2011*), or studying artificial grammatical rules elicited by structured, unintelligible strings (*Friederici et al., 2006*). Nevertheless, finding the structure in an unfolding sentence also depends on the constraints jointly placed by other linguistic properties and non-linguistic information such as the broad world knowledge (*Bever, 1970*; *Tyler and Marslen-Wilson, 1977*).

Unlike the *two-stage model* (*Frazier, 1987*; *Frazier and Rayner, 1982*) which posits an initial parsing stage relying solely on syntax, the *constraint-based* approach to sentence processing (*MacDonald et al., 1994*; *Trueswell and Tanenhaus, 1994*) proposes that speech comprehension is concurrently governed by multiple types of probabilistic constraints (e.g. syntax, semantics, world knowledge), generated by individual words as they are sequentially heard. There is no delay in the utilization of these multifaceted constraints once they become available, neither is a fixed priority assigned to one type of constraint over another; rather, it is the *interpretative coherence* of all available constraints that forms the basis for successful language comprehension (*Altmann, 1998*). Although lexical constraints of individual words can be estimated from large corpora data, it has been challenging to model the dynamic interplay between various types of linguistic and non-linguistic constraints in a specific context, especially at the sentence level and beyond.

Contemporary deep language models (DLMs) have made great strides in a wide array of natural language processing tasks, including text generation, parsing, and translation (*Vaswani, 2017*; *Devlin et al., 2019*; *Brown, 2020*; *Ouyang et al., 2022*). While current DLMs are still imperfect in terms of human-level language understanding related to reasoning and complex physical or social situations (*Bisk et al., 2020*), they are arguably valuable models of general linguistic capacities due to their ability to identify and leverage relevant statistical regularities of linguistic and non-linguistic world knowledge present in massive training data (*Linzen and Baroni, 2021*; *Pavlick, 2022*). Human language comprehension requires a contextualized integration of multifaceted constraints (*Tyler and Marslen-Wilson, 1977*; *Marslen-Wilson, 1975*; *Kuperberg, 2007*). In this regard, DLMs excel in flexible combination of different types of features (e.g. syntactic structure and semantic meaning) embedded in their rich internal representations (*Linzen and Baroni, 2021*; *Pavlick, 2022*; *Bengio et al., 2021*; *Manning et al., 2020*). Their deep contextualized representations capture the distributed regularities that jointly determine the coherent interpretation of a given sentence, providing context-dependent composition and quantitative measures of the underlying sentential structure. These properties relate back to Elman's recurrent neural network (*Elman, 1990*; *Elman, 1993*) which automatically picks up and encodes lexical syntactic/semantic information in the hidden states.

Recent studies have revealed an overall congruence between language representations in DLMs and those observed in the human brain while processing the same spoken or written input (*Schrimpf et al., 2021*; *Goldstein et al., 2022*; *Heilbron et al., 2022*; *Toneva et al., 2022*; *Caucheteux et al., 2022*; *Caucheteux and King, 2022*; *Caucheteux et al., 2023*), suggesting the potential value of DLMs as a computational tool to investigate the neural basis of language comprehension. To move beyond comparing the similarities between entire model hidden states and brain activity, probing techniques that can extract specific contents from DLMs (*Hewitt and Liang, 2019*; *Tenney, 2019*) make it possible to study the neural dynamics relevant to processing such specific information. The important advance here is that we can leverage the deep learning strengths of DLMs to create rigorously quantified models of the broader and multifaceted constraint environment in which a structured interpretation is constructed. Such models can be compared, dynamically, with more restricted and interpretable factors that capture the specific linguistic combinatorial constraints necessary for successful language comprehension.

Here, we take this approach further by designing sentences with contrasting linguistic structures and using a structural probe technique (*Hewitt and Manning, 2019*) to extract word-by-word contextualized representations of sentential structures from a widely used DLM, namely, BERT (*Devlin et al., 2019*). This provides the neurocomputational specificity required to elucidate the neural dynamics underlying the incremental construction of a structured interpretation from an unfolding spoken sentence. After a detailed evaluation of BERT structural measures according to the hypothesized *constraint-based* approach and human behavioural results, we used spatiotemporal searchlight representational similarity analysis (ssRSA) (*Kriegeskorte et al., 2008*) to test these quantitative structural

**Active interpretation**     **Passive interpretation**

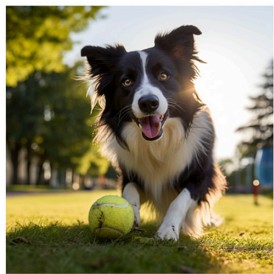

The dog **found** …
The dog **found** in the park …

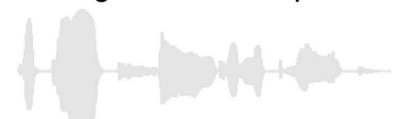

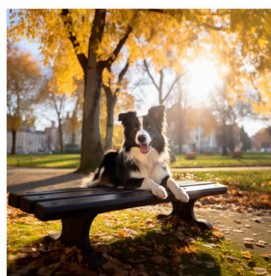

The dog **walked** …
The dog **walked** in the park …

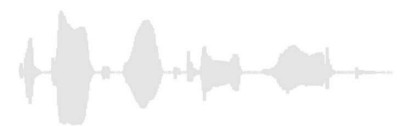

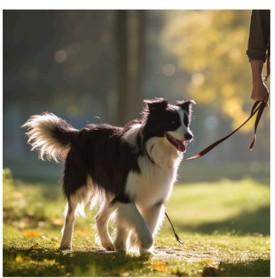

**Active index**     **Passive index**
SN agenthood x Verb1 intransitivity     SN patienthood x Verb1 transitivity

**Figure 1.** Example spoken sentence stimuli and plausible structured interpretations. The two target sentences in each set differ only in the transitivity of the first verb (Verb1). Each sentence has two possible structured interpretations before the actual main verb is presented: an active interpretation, where the subject noun (SN) performs the action, and a passive interpretation, where the SN is the recipient of the action. The interpretative preference hinges on the likelihood of the SN acting as an agent or a patient (i.e. its thematic role) in conjunction with the transitivity of Verb1. As the sentence progresses to the prepositional phrase, a combination of higher SN agenthood and greater Verb1 intransitivity (i.e. a higher active index) generally favours an active interpretation. Conversely, increased SN patienthood coupled with higher Verb1 transitivity (i.e. a higher passive index) may lead to a passive interpretation. Note that while the SN is the same for the two target sentences within the same set, it varies across different sentence sets. All images were generated using Midjourney for illustrative purposes.

measures and relevant lexical properties against source-localized EMEG data recorded while participants were listening to the same sentences. Our findings reveal how the structured interpretation of a spoken sentence is incrementally built under multifaceted probabilistic constraints in the brain.

## Results

We constructed 60 sets of sentences with varying sentential structures (see 'Materials and methods') and presented them to human listeners. We also input them word-by-word to BERT to extract incremental structural representations. These natural spoken sentences were constructed to balance off specifically linguistic constraints on interpretation against varying non-linguistic constraints as the sentence is incrementally interpreted, providing a realistic simulation of real-life language use. In each stimulus set, there are two target sentences differing only in the transitivity of the first verb (Verb1) encountered, that is, how likely it is that Verb1 takes a direct object see [a] and [b] below and *Figure 1*:

a. *The dog found in the park was covered in mud.*
b. *The dog walked in the park was covered in mud.*

In the first sentence, Verb1 (i.e. '*found*') has high transitivity (HiTrans) and strongly prefers a direct object (e.g. ball), while in the second sentence, Verb1 (i.e. '*walked*') has relatively low transitivity (LoTrans) and is often used without a following direct object. Critically, the structural interpretation of these sentences is ambiguous at the point Verb1 is encountered, and the preferred human resolution of this ambiguity depends on the real-time integration of linguistic and non-linguistic constraints as more of the sentence is heard. In the example above, the sequence "*The dog found …*" could initially have either an active interpretation – where the dog has found something – or a passive interpretation

– where the dog is found by someone (*Figure 1*). Because '*find*' is primarily a transitive verb, the human listener is likely to be biased towards an initial active interpretation. Similarly, the sequence *"The dog walked …"*, where *walk* is primarily used as an intransitive verb (without a direct object), could also bias the listener to an active interpretation, where the dog is doing the walking, rather than the less frequent passive interpretation where someone is taking the dog for a walk (i.e. walking the dog).

This initial structural interpretation up to Verb1 does not, however, just depend on linguistic knowledge such as Verb1 transitivity. It also depends on non-linguistic information, that is, how likely the subject is (or is not) to adopt the active (agent) role to perform the specified action (*Dowty, 1991*; *Marslenwilson et al., 1993*), that is, 'thematic role' properties of the subject noun. Although it could be reflected by statistical regularities in language, thematic role preference hinges more on world knowledge, plausibility, or real-world statistics. So, regardless of Verb1 transitivity, the active interpretation should be more strongly favoured in *"The **king** found/walked …"* given the higher agenthood of the '*king*' and thus the greater implausibility of a passive interpretation involving a '*king*' relative to a '*dog*'. Hence, the word-by-word interpretation of the sentential structure – and of the real-world event structure evoked by this interpretation – is determined by the constraints jointly placed by the subject noun and Verb1, which is manifested by the interpretative coherence between non-linguistic world knowledge (i.e. thematic role preference) and linguistic knowledge (i.e. verb transitivity).

As the sentence evolves, and the prepositional phrase '*in the park*' that follows Verb1 is incrementally processed, there is further modulation of the preferred interpretation, again reflecting both Verb1 transitivity and the plausibility of the event being constructed. Specifically, the passive interpretation will become more preferred in a HiTrans sentence, given the absence of an expected direct object for the highly transitive Verb1, so Verb1 tends to be interpreted as a passive verb [i.e. the head of a reduced relative clause in *"The dog **(that was) found in the park** …"*]. Conversely, in a LoTrans sentence, the active interpretation of Verb1 is strengthened by the incoming prepositional phrase, which is in accord with the verb's intransitive use and the event conjured up by the sequence of words heard so far (e.g. *"The dog walked in the park …"*). Hence, these two sentence types are likely to differ in the structural interpretation preferred by the end of the prepositional phrase. However, with the appearance of the actual main verb (e.g. '*was covered*' in the example sentences), the active interpretation of Verb1 as the main verb will be completely rejected, which resolves the potential ambiguity and confirms the passive interpretation in both HiTrans and LoTrans sentences.

In brief, understanding these complex sentences require listeners to integrate discontinuous words to solve a long-distance dependency between the subject noun and the actual main verb separated by an intervening clause. This engages the neurobiological processes of integration across different lexical constraints and multiple levels of the sentence processing system. For example, the incremental building, maintenance, and update of sentential structure over time might primarily involve activity in the fronto-temporal regions (*Friederici, 2012*), while estimating the plausibility of the event interpreted from the sentence with prior knowledge of the world may elicit neural responses in the default mode network (DMN) (*Yeshurun et al., 2021*).

## Human incremental structural interpretations

As the first step, and to quantify how the stimulus sentences exemplified a constraint-based account of incremental structural interpretation, we conducted two pre-tests where participants listened to sentence fragments, starting from sentence onset and continuing either until the end of Verb1 or to the end of the prepositional phrase (*Figure 2A*), and then produced a continuation to complete the sentence (see 'Materials and methods'). Based on the continuations provided by the listeners at these two gating points, we can infer their online structural interpretations.

In the continuations after Verb1, a direct object was more likely to be found in HiTrans sentences, indicating a transitive use of Verb1, while an opposite pattern was found for a PP continuation, indicating an intransitive use of Verb1 (*Figure 2B*). As expected, the probability of a main verb (MV) in the continuations after the prepositional phrase was lower in LoTrans sentences (*Figure 2C*), suggesting that listeners preferred the active interpretation and tended to interpret Verb1 as the main verb by the end of the prepositional phrase in LoTrans sentences, and vice versa in HiTrans sentences. Crucially, neither of the two pre-tests resulted in a complete separation between HiTrans and LoTrans sentences; instead, they were characterized by two different but overlapping probabilistic distributions. This

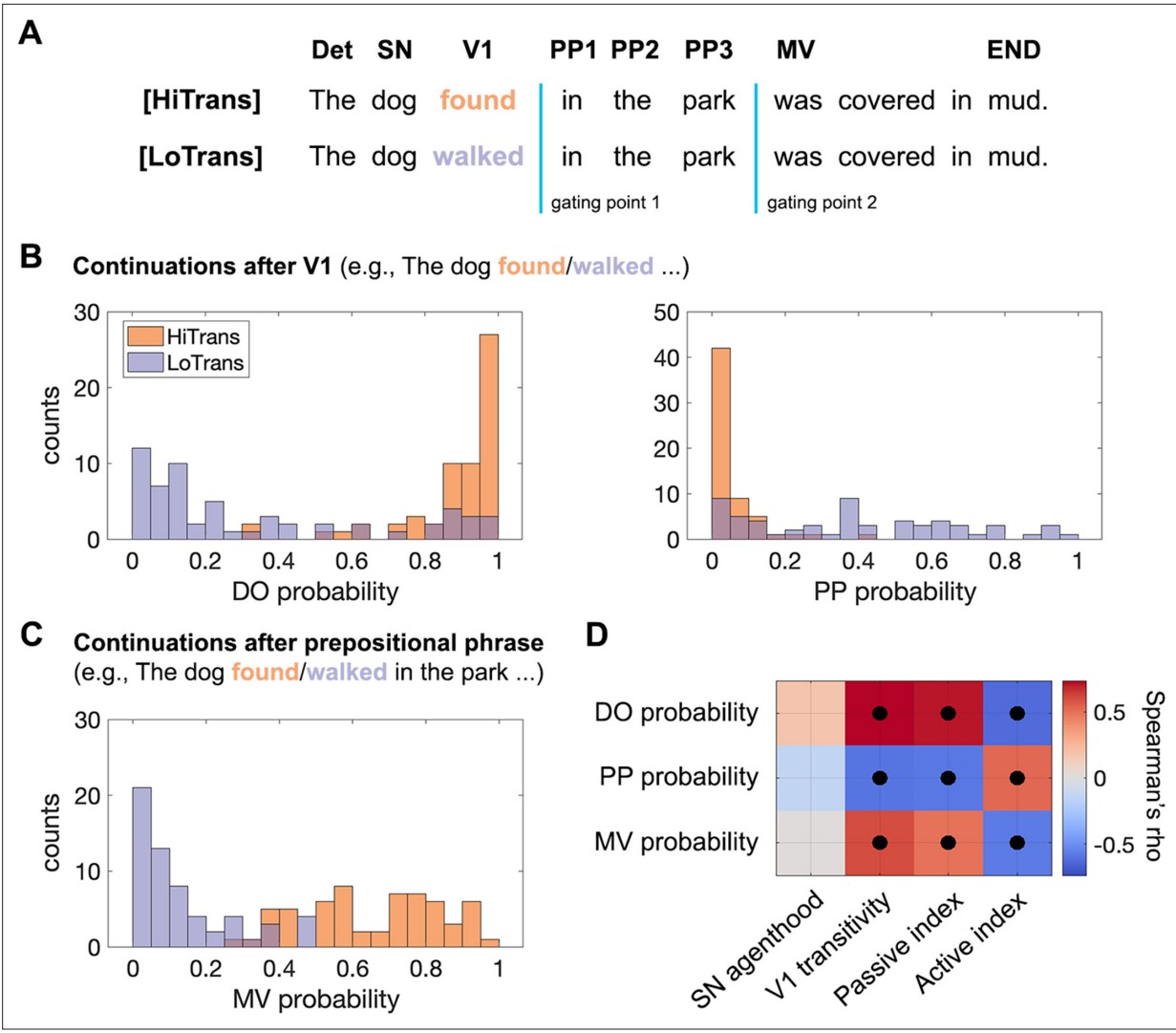

**Figure 2.** Human incremental structural interpretations derived from continuation pre-tests. (**A**) An example set of target sentences differing only in the transitivity of Verb1, HiTrans: high transitivity; LoTrans: low transitivity. Det: determiner; SN: subject noun; V1: Verb1; PP1–PP3: prepositional phrase; MV: main verb; END: the last word in the sentence. (**B**) Probability of a direct object (left) and a prepositional phrase (right) continuation after Verb1. (**C**) Probability of a main verb in the continuations after Verb1, which indicates an active interpretation. (**D**) Correlations between corpus-based lexical constraints and probabilistic interpretations in the two pre-tests (Spearman rank correlation, black dots indicate significance determined by 10,000 permutations, $P_{FDR}<0.05$ corrected).

finding suggests that passive and active interpretations varied in plausibility in each sentence type before the actual main verb was presented, reflecting the probabilistic constraints jointly placed by the combination of the specific subject noun, Verb1, and the prepositional phrase in each sentence.

To relate these human interpretative preferences to the broader landscape of distributional language data, we developed corpus-based measures of the thematic role preference of the subject noun (i.e. how likely it is interpreted as an agent that conducts an action) and the transitivity of Verb1 in each sentence, from which we derived a passive index and an active index (see 'Materials and methods'). These indices separately capture the interpretative coherence between these two lexical properties towards passive and active interpretations. Both high subject noun agenthood and low Verb1 transitivity coherently preferred an active interpretation as the prepositional phrase was heard (i.e. a high active index), and vice versa for the passive interpretation (i.e. a high passive index). In accord with the *constraint-based* hypothesis, we found that human interpretative preference for the two types of sentences was significantly correlated with the lexical constraints placed by the subject noun and Verb1 (*Figure 2D*).

## Incremental structural representations extracted from BERT

Next, we extracted structural representations at various positions in the same sentences from BERT and evaluated them according to the *constraint-based* hypothesis and human behavioural results. This evaluation is needed to motivate the use of BERT structural measures to reveal how the structured interpretation of a spoken sentence is incrementally built in the brain.

Typically, the structure of a sentence can be represented by a dependency parse tree (*MacCartney and Manning, 2006*), where words are situated at different depths given their structural dependency (*Figure 3A*). Each edge links two structurally proximate words as being the head and the dependent separately (e.g. a verb and its direct object). However, such a parse tree is context-free, that is, it only captures the syntactic relation between each pair of words and abstracts away from the specific lexical (and higher order) contents of the sentence that constrain its structural interpretation. This context-free parse depth is always the same for words at the same position in sentences with the same structured interpretation (e.g. '*found*' and '*walked*' in either of the two parse trees in *Figure 3A*).

To obtain structural measures that also encode the specific lexical contents in a sentence, we adopted a structural probing technique (*Hewitt and Manning, 2019*) to reconstruct a sentence's structure by estimating each word's parse depth based on their contextualized representations generated by BERT (see 'Materials and methods'). Note that BERT is a multi-layer DLM (24 layers in the version used in this study) which may distribute different aspects of its computational solutions over multiple layers. Accordingly, we trained a structural probing model for each layer and selected the one with the most accurate structural representations while also including its neighbouring layers to cover relevant upstream and downstream information. Following this strategy, we used the BERT structural measures obtained from layers 12–16 with the best performance achieved in layer 14 (see *Figure 3— figure supplement 1* and 'Materials and methods').

We input each sentence word-by-word to the trained BERT structural probing models, focusing on the incremental structural representation being built as it progressed from Verb1 to the main verb (see examples in *Figure 3B and C*). Note that we defined the first word after the prepositional phrase as the main verb since its appearance is sufficient to resolve the intended structure where Verb1 is a passive verb. We found that, for each type of sentences, the BERT parse depth of words at the same position formed a distribution ranging around the corresponding context-free parse depths in either the passive or the active interpretation (see *Figure 3—figure supplement 2*), suggesting a word-specific rather than position-specific structural representation. In addition, we quantified the contributions of words at different positions to the variances encoded in BERT parse depth vectors. Our analysis revealed that content words contributed significantly more than function words (i.e. the determiners) (see *Figure 3—figure supplement 3*).

Then we visualized BERT's word-by-word structural measures, focusing on the dependency between the subject noun and Verb1 that is core to the current interpretation of the sentence – whether the subject noun is the agent or the patient of Verb1. To this end, we built a three-dimensional (3D) vector including BERT parse depths of the first three words up to Verb1 for each sentence (e.g. *"The dog found …"*). This 3D vector was updated incrementally with each additional word in the input, thereby capturing the dynamic interpretation of the structural dependency between the subject noun and Verb1, influenced by the context provided by subsequent words in a specific sentence. Like the probabilistic interpretation found within each type of sentences in human listeners, trajectories of individual HiTrans and LoTrans sentences are considerably distributed and intertwined (see the upper panel of *Figure 3D*), implying that BERT structural interpretations are sensitive to the idiosyncratic contents in each sentence.

To make sense of these trajectories, we also vectorized the context-free parse depth of the first three words indicating passive and active interpretations (*Figure 3A*) separately and located them in the 3D vector space as landmarks (hollow triangle and circle in *Figure 3D*), so that the plausibility of either interpretation can be estimated by a sentence's distance from its landmark. As shown by the trajectories of the median BERT parse depth of the two sentence types (see the lower panel of *Figure 3D*), in general, HiTrans sentences continuously moved towards the passive interpretation landmark after Verb1, with a significant change of distances detected at the main verb (see the orange bars in *Figure 3E*). LoTrans sentences started by approaching the active interpretation landmark but were reorientated to the passive counterpart with the appearance of the actual main verb, with significant changes of distances detected at both Verb1 and main verb (see the purple bars in *Figure 3E*).

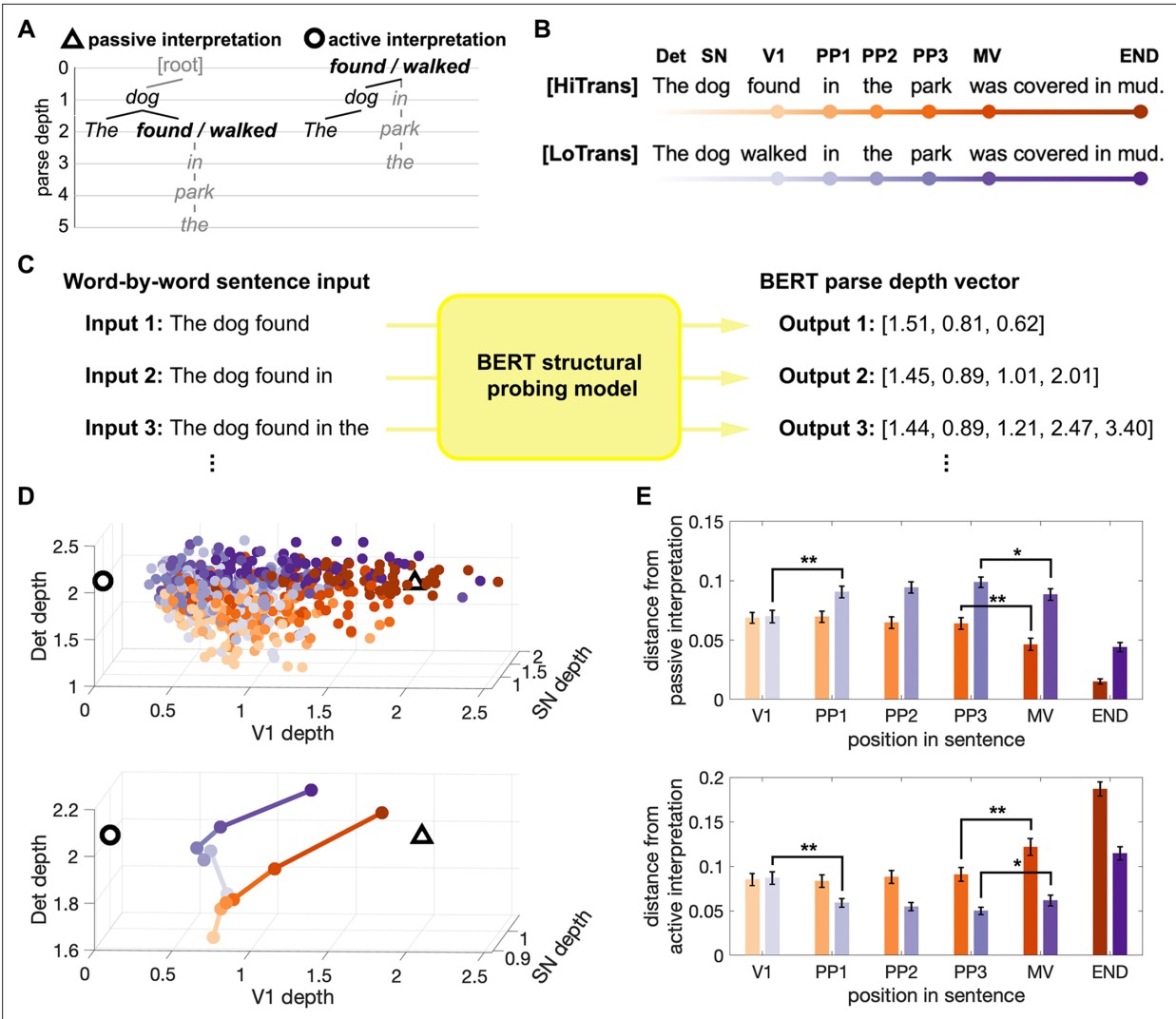

**Figure 3.** Incremental interpretation of sentential structure by BERT. (**A**) Context-free dependency parse trees of two plausible structural interpretations. Left: passive interpretation where V1 is the head of a reduced relative clause. Right:aActive interpretation where V1 is the main verb. (**B**) Incremental input to BERT structural probing model, with the lightness of dots encoding different positions in the target sentences. Det: determiner; SN: subject noun; V1: Verb1; PP1–PP3: prepositional phrase; MV: main verb; END: the last word in the sentence. (**C**) BERT structural probing model is trained to output a parse depth vector, representing the parse depths of all the words in the sentence input. The BERT parse depth for a specific word is updated incrementally as the sentence unfolds word-by-word. In this example, the parse depth of '*found*' increases with the presence of the prepositional phrase, indicating an increased preference for the passive interpretation according to the context-free parse depths in (**A**). (**D**) Incremental interpretation of the dependency between SN and V1 in the model space consisting of the parse depth of Det, SN, and V1. Upper: Each coloured circle represents the parse depth vector up to V1 derived at a certain position in the sentence (with the same colour scheme as in **B**). The hollow triangle and circle represent the context-free dependency parse vectors for passive and active interpretations in (**A**). Lower: incremental interpretation of the two target sentence types represented by the trajectories of median parse depth. (**E**) Distance from passive and active landmarks in the model space as the sentence unfolds (between each coloured circle and the two landmarks in the upper panel of **D**) (two-tailed two-sample *t*-test, *p<0.05, **p<0.001, n = 60 for both HiTrans and LoTrans sentences, error bars represent SEM).

The online version of this article includes the following figure supplement(s) for figure 3:

**Figure supplement 1.** Performance of structural probing models trained on different BERT layers.

**Figure supplement 2.** BERT structural representations of incremental sentence inputs.

**Figure supplement 3.** Contribution of words at different positions in BERT parse depth vectors.

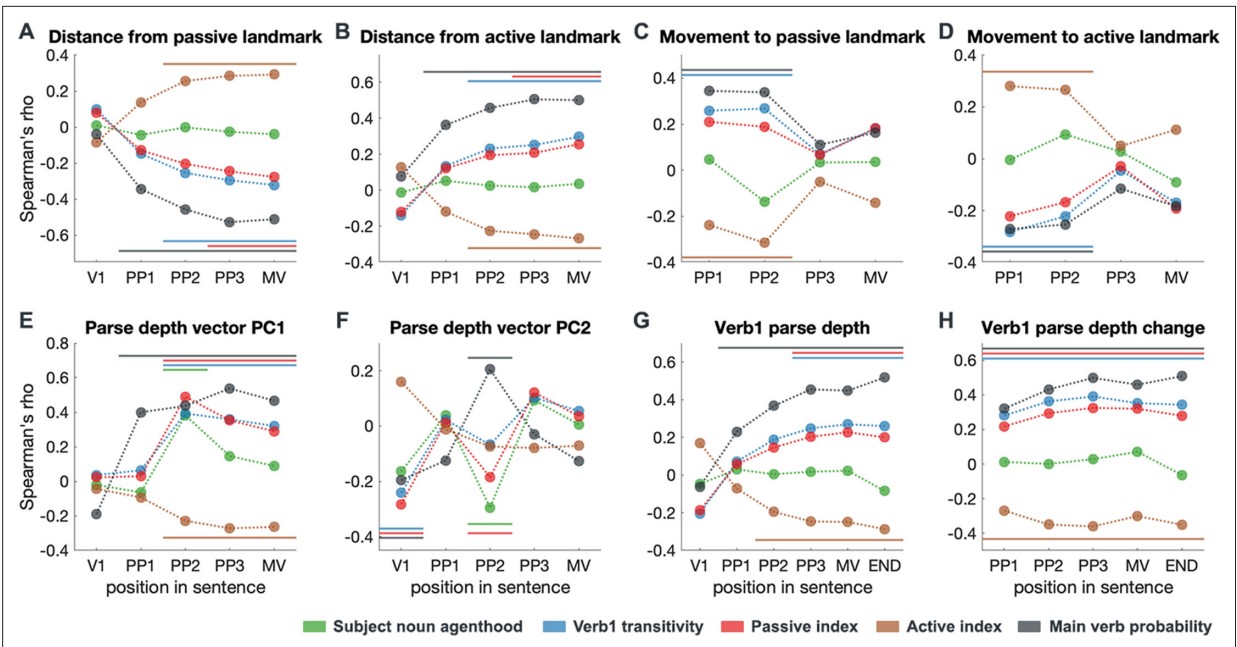

**Figure 4.** Correlation between incremental BERT structural measures and explanatory variables. BERT structural measures include (**A, B**) BERT interpretative mismatch represented by each sentence's distance from the two landmarks in model space (*Figure 3D*); (**C, D**) dynamic updates of BERT interpretative mismatch represented by each sentence's movement to the two landmarks; (**E, F**) overall structural representations captured by the first two principal components (i.e. PC1 and PC2) of BERT parse depth vectors; (**G, H**) BERT Verb1 (V1) parse depth and its dynamic updates. Explanatory variables include lexical constraints derived from massive corpora and the main verb probability derived from the continuation pre-test (Spearman correlation, permutation test, $P_{FDR}<0.05$, multiple comparisons corrected for all BERT layers, results shown here are based on layer 14, see *Figure 4— figure supplements 1–3* for the results of all layers, see *Figure 7—figure supplement 1* for the dynamic change of Verb1 parse depth); PP1–PP3: prepositional phrase; MV: main verb; END: the last word in the sentence.

The online version of this article includes the following figure supplement(s) for figure 4:

**Figure supplement 1.** Correlation between BERT structural interpretations and explanatory variables.

**Figure supplement 2.** Correlation between the principal components (PCs) of BERT parse depth vectors and explanatory variables.

**Figure supplement 3.** Correlation between BERT parse depth of individual words and explanatory variables.

These results resemble the pattern of human interpretative preference observed in the continuation pre-tests (*Figure 2C*), where the passive and active interpretations were separately preferred in HiTrans and LoTrans sentences by the end of the prepositional phrase in a probabilistic manner, before the passive interpretation was established with the appearance of the actual main verb.

## BERT structural measures are correlated with constraints driving human interpretation

To further assess whether BERT's preferences for structural interpretation align with the constraints considered by human listeners during speech comprehension, we correlated BERT structural measures with relevant corpus-based measures and human behavioural data (see 'Materials and methods).

We first focused on BERT's interpretative mismatch quantified as the distance between an unfolding sentence and each of the two landmarks in the model space, which was dynamically updated as the sentence unfolded (*Figure 3D*). Consistently, from the incoming prepositional phrase to the main verb, sentences that are closer to the passive landmark in the model space have higher Verb1 transitivity, a higher passive index but a lower active index, while sentences closer to the active interpretation landmark exhibited higher active index and lower passive index (*Figure 4A and B*). Moreover, at the beginning of the prepositional phrase, the change of distance towards either interpretation landmark between two consecutive words is also correlated with these constraints (*Figure 4C and D*), suggesting an immediate update in the structural interpretation in combination with the accumulated constraints from the preceding subject noun and Verb1.

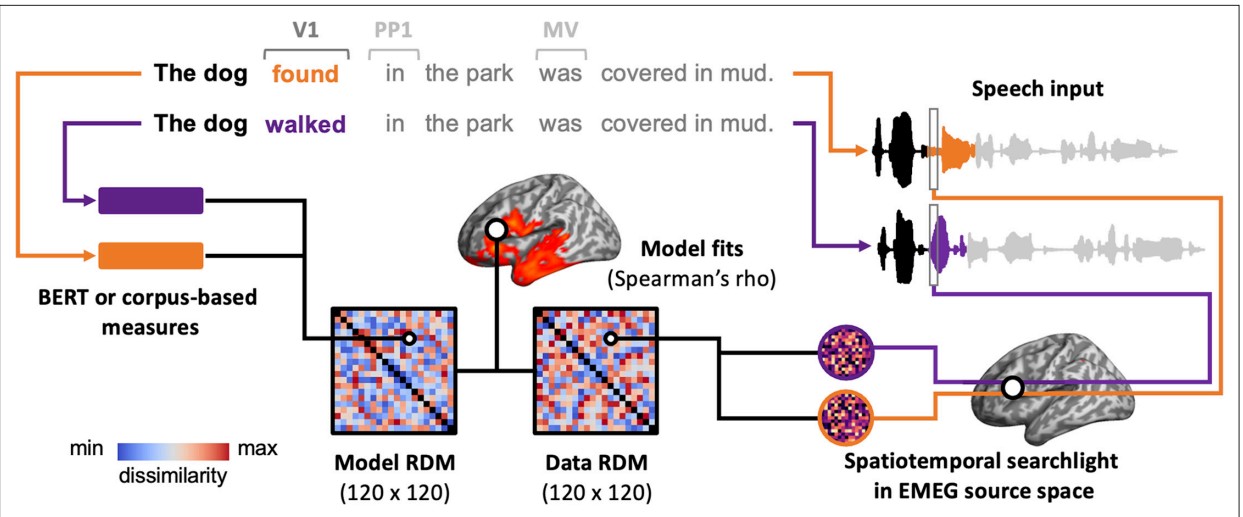

**Figure 5.** Illustration of the pipeline for spatiotemporal searchlight representational similarity analysis (ssRSA). For each pair of sentences, we extract their BERT or corpus-based measures and calculate the dissimilarity between these measures, resulting in a model representational dissimilarity matrix (RDM). Meanwhile, we also extract the neural activity recorded while participants are listening to these sentences and calculate their dissimilarity to create a data RDM. Specifically, we use a spatiotemporal searchlight in EMEG source space, which moves across the brain and captures the neural activity within a 10-mm-radius sphere over a 60 ms sliding time window. By correlating the model RDM with data RDMs from all spatiotemporal searchlights, we can identify whether, and if so, when and where the brain represents the information captured by the model RDM. The ssRSA is conducted in V1, PP1, and MV epochs, respectively, with HiTrans and LoTrans sentences combined as one group (i.e. 120 sentences in total).

Moreover, we found that both the incremental BERT parse depth vectors as a whole (which are captured by their principal components) and the BERT parse depth of Verb1 (which is the most indicative marker of the interpretation preferred) are correlated with the constraints placed by the subject noun and Verb1 (*Figure 4E–H*). Moreover, the significant effects consistently found as the sentence unfolds suggest that properties of preceding words are used to constrain the interpretation of the upcoming input, which is key to resolving discontinuous structural dependencies. In addition, we found that BERT structural interpretations were also correlated with the main verb probability in the continuation pre-test which directly reflects human interpretation preference (black bars in *Figure 4*).

Overall, these results illustrated, at which position in a sentence, relevant lexical constraints started being encoded by BERT, which also validated the contextualized BERT structural measures in terms of the *constraint-based* hypothesis and human behavioural results, and motivated the use of them to probe the neural processes involved during the incremental structural interpretation of spoken sentences.

## Neural dynamics of incremental structural interpretation

To study how the structured interpretation of a spoken sentence is built word-by-word in the brain, we used ssRSA to test the incremental BERT structural measures in source-localized EMEG collected when the same sentences were delivered to human listeners (see *Figure 5* for the pipeline of ssRSA). This combination of methods gains improved neurocomputational specificity by probing the spatio-temporally resolved neural activity with detailed structural representations rather than the entire hidden states of BERT. We compared the representational geometry of BERT structural measures with that of neural responses inside a spatiotemporal searchlight moving across the cortical surface, significant model fits showed when and where the incremental structural interpretations or relevant lexical constraints emerge and update in the brain. Given the probabilistic interpretations in BERT and human listeners reported above, we combined HiTrans and LoTrans sentences as one group to increase the range of pair-wise dissimilarity to be modelled in ssRSA.

We began with the BERT parse depth vector containing the parse depth of each word in an incremental input, providing a dynamic structural representation updated as the sentence unfolded. Then, we tested the interpretative mismatch between the incremental BERT parse depth vector and the corresponding context-free parse depth vector for the passive or the active interpretation. The degree of this mismatch is proportional to the evidence for or against the two interpretations, that is, the

smaller the distance, the more positively loaded this interpretation. Besides these two measures based on the entire incremental input, we also focused on Verb1 since the potential structural ambiguity lies in whether Verb1 is interpreted as a passive verb or the main verb. Given the context-free parse depth of Verb1 that is 2 in the passive interpretation and 0 in the active interpretation (*Figure 3A*), with each incoming later word, an increased BERT Verb1 parse depth towards 2 or a decreased value towards 0 reflects separately the preference biased to a passive or an active interpretation (*Figure 7—figure supplement 1*). All quantitative measures tested in ssRSA are summarized in *Supplementary file 1*.

For the listener's neural activity, we focused on three critical epochs in each sentence: (a) Verb1 – when its structural dependency with the preceding subject noun was initially established despite potential ambiguity; (b) the preposition – when the initial structural interpretation started being

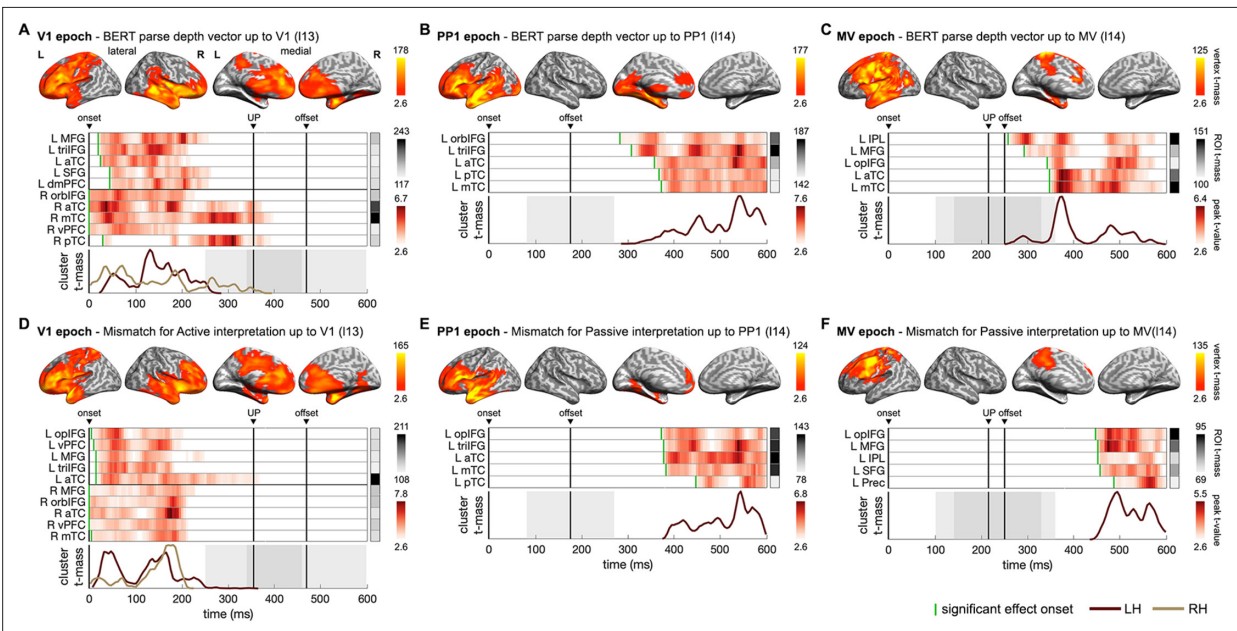

**Figure 6.** Neural dynamics underpinning the emerging structure and interpretation of an unfolding sentence. (**A–C**) Spatiotemporal searchlight representational similarity analysis (ssRSA) results of BERT parse depth vector up to Verb1 (V1), the preposition (PP1), and the main verb (MV) in epochs separately time-locked to their onsets. (**D–F**) ssRSA results of the mismatch for the preferred structural interpretation (the specific BERT layer from which BERT structural measures were derived is denoted in parentheses). From top to bottom in each panel: vertex *t*-mass (each vertex's summed *t*-value during its significant period); heatmap of time series of region of interest (ROI) peak *t*-value (the highest *t*-value in an ROI at each time point) with a green bar indicating effect onset and ROI *t*-mass (each ROI's summed mean *t*-value during its significant period); cluster *t*-mass time series (summed *t*-value of all the significant vertices of a cluster at each time point) (cluster-based permutation test, vertex-wise p<0.01, cluster-wise p<0.05 in **A–E**; marginal significance in **F** with cluster-wise p=0.06). Solid vertical lines indicate the timings of onset, average uniqueness point (UP), and average offset of the word time-locked in the epoch with grey shades indicating the range of 1 SD. LH/RH: left/right hemisphere. See *Supplementary file 2* for full anatomical labels. See *Figure 6—figure supplement 1* for Spearman's rho time series of ROIs in individual participants, and *Figure 6—figure supplement 2* for the significant results of other BERT layers in the MV epoch.

The online version of this article includes the following figure supplement(s) for figure 6:

**Figure supplement 1.** Spearman's rho time series of ROIs across individual participants and their mean (with SEM) for BERT parse depth vector and its mismatch for active and passive interpretations in V1, PP1 and MV epochs.

**Figure supplement 2.** Spatiotemporal searchlight representational similarity analysis (ssRSA) results of BERT structural measures in the main verb (MV) epoch.

**Figure supplement 3.** Spatiotemporal searchlight representational similarity analysis (ssRSA) results of BERT structural measures in the Verb1 (V1) epoch.

**Figure supplement 4.** Comparison between the representational similarity analysis (RSA) model fits of BERT structural metrics and behaviour-/ corpus-based metrics in the Verb1 (V1) epoch.

**Figure supplement 5.** Comparison between the representational similarity analysis (RSA) model fits of BERT structural metrics and behaviour-/corpus-based metrics in the preposition (PP1) epoch.

**Figure supplement 6.** Comparison between the representational similarity analysis (RSA) model fits of BERT structural metrics and behaviour-/ corpus-based metrics in the main verb (MV) epoch.

updated, to be either strengthened or weakened by the incoming preposition phrase; and (c) main verb – when the intended passive interpretation was finally confirmed. We aligned the continuous EMEG data to the onset of Verb1, the preposition, and the main verb respectively and obtained three 600 ms epochs.

We found that the incremental BERT parse depth vectors exhibited significant fits to brain activity consistently in all three epochs as the corresponding word was being heard at that time (*Figure 6A–C*). In Verb1 epoch, effects in bilateral frontal and anterior-to-middle temporal regions started immediately from Verb1 onset and continued until the uniqueness point – the point at which the word has been uniquely identified. These early effects could be due to the different subject nouns included in the BERT parse depth vectors. While the BERT parse depth of Verb1 per se showed similar effects but with greater duration which peaked exactly at Verb1 uniqueness point (*Figure 6—figure supplement 3*). As the sentence unfolded, effects of BERT parse depth vectors were found in the left fronto-temporal regions in the two later epochs, starting after the recognition of the preposition or the main verb separately.

Turning to the interpretative mismatch for the two possible interpretations, we only observed significant effects of the mismatch for active interpretation in Verb1 epoch (*Figure 6D*). However, it was the mismatch for passive interpretation that fitted brain activity in the preposition and main verb epochs (*Figure 6E and F*, marginal significance in main verb epoch with cluster-wise p=0.06). These results suggest that listeners, in general, tended to have an initial preference for an active interpretation (even before the recognition of Verb1) but might start favouring a passive interpretation when the prepositional phrase began to be heard. This finding is consistent with the tendency to process the first noun encountered at the beginning of a sentence as the agent (*Bever, 1970*; *Jackendoff, 2002*; *Mahowald et al., 2023*). Note that our approach does not constitute a direct test for the hypothesis of parallel parsing, as we did not uncover evidence supporting the maintenance of parallel representations of different syntactic structures in the brain; rather, we only found one preferred structure in each epoch.

Effects of the BERT parse depth vectors and those of the interpretative mismatch for the preferred structural interpretation have substantial overlaps in terms of their spatiotemporal patterns in the brain, characterized primarily by a transition from bilateral to left-lateralized fronto-temporal regions as the sentence unfolds. Across the three epochs, the most sustained effects were observed in the left inferior frontal gyrus (IFG) and the anterior temporal lobe (ATL). Notably, with the identification of the actual main verb, effects of the eventually resolved structure also involved regions in the left prefrontal and inferior parietal regions (*Figure 6C*) which belong to the multiple-demand network (*Duncan, 2010*). The involvement of the prefrontal regions could be indicative of the varying working memory demands (e.g. the different number of open nodes in the sentence structures for active and passive interpretations before the actual main verb is recognized) for building the structure of the unfolding sentence (*Nelson et al., 2017*; *Pallier et al., 2011*).

## Structural ambiguity resolution probed using BERT Verb1 parse depth

As mentioned above, the potential ambiguity between a passive and an active interpretation centres around whether Verb1 is considered as a passive verb or the main verb, which is resolved upon the appearance of the actual main verb. We probed how this is implemented in the brain using the dynamic BERT parse depth of Verb1. Specifically, the cognitive demands required by this resolution process can be characterized by the change between the updated BERT parse depth of Verb1 when the actual main verb is presented and its initial value when Verb1 is first encountered (see *Figure 7—figure supplement 1* for the dynamic change of BERT Verb1 parse depth).

We first tested the change of Verb1 parse depth in the main verb epoch. Significant fits to brain activity emerged in the left posterior temporal and inferior parietal regions upon the main verb uniqueness point, and then extended to more anterior temporal regions (*Figure 7A*). After the main verb offset, the declining effects of the Verb1 parse depth change in the left anterior temporal region seamlessly overlapped with the arising effects of the updated Verb1 parse depth (*Figure 7B and C*). These results indicate that the recognition of the actual main verb immediately triggered an update of the previous interpretation of Verb1, with the resolved interpretation emerged in the left temporal lobe and was later delivered to the right posterior temporal and parietal areas. It is also worth noting that the left hippocampus was activated for both measures of Verb1 parse depth after the actual main

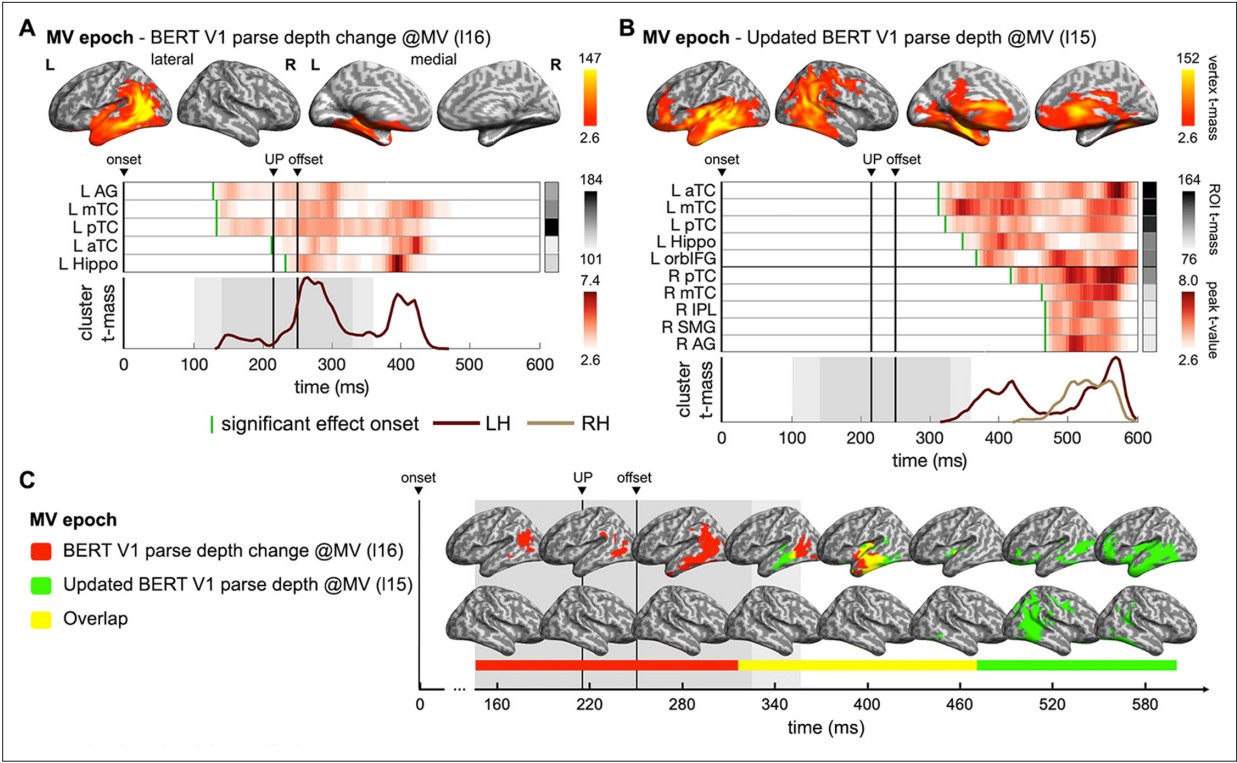

**Figure 7.** Neural dynamics updating the incremental structural interpretation. (**A**) Spatiotemporal searchlight representational similarity analysis (ssRSA) results of BERT Verb1 (V1) parse depth change at the main verb (MV) relative to the parse depth of V1 when it is first encountered. (**B**) ssRSA results of the updated BERT V1 parse depth when the input sentence reaches the MV. (**C**) Spatiotemporal overlap between the effects in (**A**) and (**B**). (cluster-based permutation test, vertex-wise p<0.01, cluster-wise p<0.05). See *Figure 7—figure supplement 2* for Spearman's rho time series of ROIs in individual participants.

The online version of this article includes the following figure supplement(s) for figure 7:

**Figure supplement 1.** The dynamic change of BERT Verb1 (V1) parse depth in unfolding sentences.

**Figure supplement 2.** Spearman's rho time series of ROIs across individual participants and their mean (with SEM) for BERT V1 parse depth change and the updated BERT V1 parse depth in the MV epoch.

verb is recognized, suggesting that the episodic memory of experienced events might contribute to the updating of structural interpretations (*Bicknell et al., 2010*; *Altmann and Ekves, 2019*; *Metu-salem et al., 2012*). These results address the dynamic update of structured interpretation by focusing on the BERT parse depth of Verb1, which complements those of the interpretative mismatch based on the incremental BERT parse depth vector incorporating constraints of all the words heard so far (*Figure 6F*).

## Emergent structural interpretations driven by multifaceted constraints in the brain

Next, we further asked how the multifaceted constraints, considered by human listeners and encoded in BERT parse depths, drive the structured interpretation in the brain? When and where in the brain do these constraints emerge? How are their neural effects related to those of the final resolved senten-tial structure? To address these questions, we first tested the subject noun thematic role properties obtained from corpus data. Significant effects of agenthood and patienthood were found in the prep-osition epoch (*Figure 8A*) and in the main verb epoch (*Figure 8B*) separately. Notably, effects of the subject noun itself preceded those of incremental BERT parse depth vectors modelling the sentence fragments in the same epoch (compare *Figure 8A* with *Figure 6B* and *Figure 8B* with *Figure 6C*). These findings indicate that subject noun thematic role might be evaluated before building the overall structural interpretation of the utterance delivered so far. Specifically, the initial preference for an active interpretation during Verb1, while present as the preposition started (i.e. subject noun

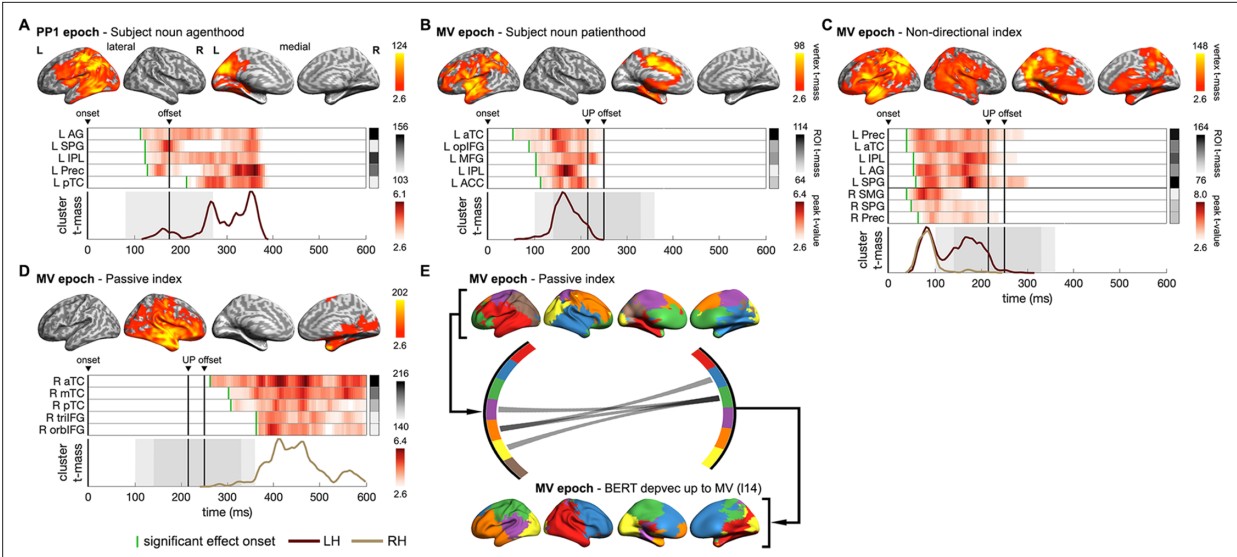

**Figure 8.** Neural dynamics of multifaceted probabilistic constraints underpinning incremental structural interpretations. (**A, B**) Spatiotemporal searchlight representational similarity analysis (ssRSA) results of subject noun (SN) agenthood and SN patienthood (i.e. plausibility of SN being the agent or the patient of Verb1 [V1]) in PP1 and main verb (MV) epochs separately. (**C**) ssRSA results of non-directional index (i.e. interpretative coherence between SN and V1 regardless of the structure preferred) in MV epoch. (**D**) ssRSA results of passive index (i.e. interpretative coherence for the passive interpretation) in MV epoch. (**E**) Influence of the passive interpretative coherence on the emerging sentential structure in MV epoch revealed by the Granger causal analysis (GCA) based on the non-negative matrix factorization (NMF) components of whole-brain ssRSA results (see *Figure 8—figure supplement 1* for more details). (**A–D**) Cluster-based permutation test, vertex-wise p<0.01, cluster-wise p<0.05; (**E**) permutation test P$_{FDR}$<0.05. See *Figure 8—figure supplement 2* for Spearman's rho time series of ROIs in individual participants.

The online version of this article includes the following figure supplement(s) for figure 8:

**Figure supplement 1.** Directional relationship between multifaceted constraints and structured interpretation in the brain.

**Figure supplement 2.** Spearman's rho time-series of ROIs across individual participants and their mean (with SEM) for corpus-based measures in PP1 and MV epochs.

**Figure supplement 3.** Illustration of directionality of dissimilarity geometry in the representational dissimilarity matrix (RDM) based on a ratio measure.

agenthood in *Figure 8A*), was superseded by the preference for a passive interpretation as the rest of the prepositional phrase (*Figure 6A*) and the main verb (*Figure 6B*) were heard.

Despite being jointly constrained by subject noun thematic role preference and Verb1 transitivity in a probabilistic manner, the structural interpretation temporarily held just before the recognition of the actual main verb could differ across sentences (e.g. passive interpretation in "*The dog found in the park …*" and active interpretation in "*The dog walked in the park …*"). Therefore, in contrast to the passive or active index specialized for one particular structural interpretation, we constructed a non-directional index that merely quantifies the degree of interpretative coherence for one interpretation, whether passive or active (see 'Materials and methods' for details). Thus, a higher value only indicates greater interpretative coherence between the subject noun and Verb1 regardless of which interpretation is preferred.

Effects of this non-directional measure of interpretative coherence appeared very soon after the main verb onset in both hemispheres and lasted till its offset (*Figure 8C*), suggesting an immediate evaluation of the previously integrated constraints from the subject noun and Verb1 when a listener realized that the sentence had not finished yet. Moreover, these effects roughly co-occurred with the effects of subject noun patienthood (compare *Figure 8B and C*), indicating that a patient role for the subject noun was considered as the main verb was being recognized. Intriguingly, the most sustained regions associated with this non-directional index, including the left ATL, angular gyrus (AG), and precuneus, are also the classical areas of the DMN. This finding is consistent with recent claims that the DMN integrates external input with internal prior knowledge to make sense of an input stimulus such as speech (*Yeshurun et al., 2021*). In particular, precuneus and AG have been found to be involved in building thematic relationships and event structures (*Baldassano et al., 2017*; *Humphreys et al., 2021*).

Following the declining effects of the non-directional index upon the recognition of the main verb, we found significant effects of the passive index in right anterior fronto-temporal regions (*Figure 8D*), suggesting that the intended passive interpretation was eventually established in all sentences. Previous studies have revealed that the relatively narrow sentence-specific information and the broad world knowledge are processed in the left and right hemispheres separately (*Jung-Beeman, 2005*; *Metusalem et al., 2016*; *Troyer et al., 2022*). Relevant to this, in the main verb epoch, we found effects of the BERT parse depth vector and those of the passive index in the left and right hemispheres, respectively, arising almost at the same time as the main verb was recognized (compare *Figure 6C* and *Figure 8D*). Therefore, a critical question is whether and how the online structural interpretation of a specific sentence is facilitated by the interpretative coherence conjured up from lexical constraints that also depend on broad world knowledge (e.g. thematic role).

To address this question, we adopted non-negative matrix factorization to decompose the whole-brain ssRSA fits of the passive index and the BERT parse depth vector found in the main verb epoch into two sets of components given their temporal synchronizations (see 'Materials and methods'). We then conducted multivariate Granger causality analyses (GCA) to infer directed connections among them. We found only GC connections from the components of passive index to those of BERT parse depth vector (*Figure 8E*). Specifically, we identified information flows from the right hemisphere components of the passive index to the left hemisphere components of BERT parse depth vector, suggesting that a sentence's structure represented in the left hemisphere might be influenced by the coarse estimate of the event plausibility concurrently determined by broad world knowledge in the right hemisphere (*Jung-Beeman, 2005*; see *Figure 8—figure supplement 2* for more details).

## Comparisons between BERT and corpus/behavioural measures in fitting neural activity

To directly assess the performance of BERT structural measures with that of traditional measures extracted from corpus or behavioural data in fitting listeners' neural activity, we also conducted ssRSA with model RDMs of corpus-based or behavioural measures. In the Verb1 epoch, we tested Verb1 transitivity obtained from either corpus data or human continuations; however, neither of them exhibited significant model fits, which contrasted with the pronounced effects of BERT Verb1 parse depth (*Figure 6—figure supplement 4*). Similarly, in the PP1 and MV epochs, the probabilities of PP and MV continuations, as determined from behavioural data, did not show any significant model fits (*Figure 6—figure supplements 5 and 6*). Furthermore, the effects of BERT parse depth vector in these two epochs (*Figure 6B and C*) remained largely unchanged after controlling for the variance explained by the behavioural measures. These findings suggest that BERT structural measures, compared to corpus-based and behavioural measures, are better at fitting the neural dynamics during incremental speech comprehension. This might be attributed to the capacity of DLMs to capture more nuanced and contextually rich representations (*Linzen and Baroni, 2021*; *Pavlick, 2022*).

## Discussion

In this study, we investigated the neural dynamics involved in constructing structured interpretations from speech. We combined spatiotemporally resolved brain activity of human listeners, quantitative structural representations derived from a DLM (i.e. BERT), and corpus-based and behavioural measures. Our study revealed the emergence and update of a structured interpretation, jointly constrained by different lexical properties related to both linguistic and non-linguistic world knowledge, in an extensive set of brain regions beyond the core fronto-temporal language network. Specifically, our results show (1) a shift from bi-hemispheric lateral frontal-temporal regions to left-lateralized regions in representing the current structured interpretation as a sentence unfolds, (2) a pattern of sequential activations in the left lateral temporal regions, updating the structured interpretation as syntactic ambiguity is resolved, and (3) the influence of lexical interpretative coherence activated in the right hemisphere over the resolved sentence structure represented in the left hemisphere. These findings provide empirical evidence for the *constraint-based* approach to sentence processing and deepen the understanding of specific spatiotemporal patterning and neuro-computational properties underpinning incremental speech comprehension.

Using artificial neural networks (ANNs) to study the neural substrates of human cognition complements the long-time pursuit of generative rules and interpretable models (*Kriegeskorte and Douglas, 2018*). ANNs have informed our understanding of various cognitive processes in the brain by providing quantifiable predictions that aim to connect behaviours and relevant neural activity (*Yamins and DiCarlo, 2016*; *Rabovsky et al., 2018*; *Donhauser and Baillet, 2020*; *Yang et al., 2019*; *Kietzmann et al., 2019*; *Bao et al., 2020*; *Sheahan et al., 2021*; *Giordano et al., 2023*; *Doerig et al., 2023*). This is crucial for quantifying the outcome of complex, interrelated constraints that arise in specific contexts, such as spoken sentences, and constructing the representational geometry to be probed in the brain. Where DLMs are concerned, recent studies have systematically compared the internal representations of DLMs to those observed in the human brain during language processing, which highlights the importance of predictive coding and contextual information (*Schrimpf et al., 2021*; *Goldstein et al., 2022*; *Heilbron et al., 2022*; *Toneva et al., 2022*; *Caucheteux et al., 2022*; *Caucheteux and King, 2022*; *Caucheteux et al., 2023*). Furthermore, these studies have motivated the use of DLMs as a computational tool, or hypothesis, to study the neural substrates of language.

Here we asked a more specific question, that is, how a sequence of spoken words is incrementally structured and coherently interpreted in the brain? Our goal was to use quantitative measures of sentence structure that capture the interplay between different types of constraints that simultaneously influence this process. As a potential solution, we extracted detailed structural measures specific to the contents in each sentence from the hidden states of BERT, which was trained on massive corpora from real-life language use. Although DLMs such as BERT are not specifically designed to parse sentences, they can learn from training corpora the multi-dimensional properties related to sentence structure and dependency (*Manning et al., 2020*). In line with this, our analyses confirmed that BERT structural measures incorporate relevant lexical constraints and that they exhibit both behavioural and neural alignments with human listeners.

Taking advantage of the contextualized BERT structural measures, our ssRSA results provide neural evidence for the construction of a coherent interpretation driven by the interaction between linguistic and non-linguistic knowledge evoked by individual words as they are heard sequentially in a spoken sentence. Specifically, neural representations of an unfolding sentence's structure initially emerged in bilateral fronto-temporal regions and became left-lateralized when more complex syntactic properties, rather than canonical linear adjacency, were considered to build a structured interpretation (e.g. beyond Verb1 in our stimulus sentences). Meanwhile, we found right-hemisphere activations associated with broad world knowledge, which is essential for understanding the intended meaning conveyed by the speaker (*Bicknell et al., 2010*). In addition to the core fronto-temporal language network, we found that the multiple-demand network and the DMN were also involved during online construction of structured interpretations, which may reflect additional cognitive demands for resolving potential structural ambiguity and evaluating the plausibility of underlying events (*Smallwood et al., 2021*).

Moreover, our results show that, compared to corpus-based and behavioural measures, BERT structural measures are more effective in fitting listeners' neural activity, possibly due to its advanced ability in modelling specific contexts within each sentence. Nevertheless, it is important to recognize the important role of corpus-based and behavioural measures as explanatory variables. They are crucial not only in interpreting BERT measures but also in understanding their alignment with listeners' neural activity. This includes, for instance, the temporal sequence of activations of key lexical constraints and the emerging structure of a sentence (e.g. effects of subject noun patienthood leading those of BERT parse depth vector in the MV epoch, as seen in *Figures 8B* and *6C*) and the spatial distribution of their model fits in the brain (e.g. contrasting model fits of passive index and BERT parse depth vector in the MV epoch across different hemispheres, as shown in *Figures 8D* and *6C*). Such an integrative approach allows for a more comprehensive understanding of the complex mental processes underpinning speech comprehension, which takes advantages of the interpretability of traditional measures and the deep contextualized representations of DLMs.

There are two points to note about the use of BERT. Firstly, unlike autoregressive DLMs trained using left-to-right attention and the next-word prediction task, BERT is trained to predict masked words in a sentence with a bidirectional attention mechanism. The additional right-to-left attention provides updated representations of preceding words every time an incoming word is added to the input (e.g. representation of '*dog*' in "*The dog …*" is different from that in "*The dog found …*"). This

feature of BERT is useful for tracking the dynamic change of the representation of a specific word as its context evolves (e.g. Verb1 in this study), particularly in sentences with structural ambiguity. Although autoregressive DLMs also update hidden states as the input unfolds and could be used to study complex sentential structures (*Jurayj et al., 2022*), the updated contextual effects are reflected in the hidden states of the right-most incoming word, while those of the preceding words on the left remain unchanged (i.e. the representation of '*dog*' is the same in *"The dog …"* and *"The dog found …"*). This is different from BERT, where the updated contextual effects are reflected in the hidden states of all preceding words.

Secondly, although we input each sentence word-by-word to BERT, however, unlike human listeners or recurrent neural networks, BERT process two consecutive inputs (e.g. *"The dog …"* and *"The dog found …"*) independently, and there is no direct relationship between the hidden states of these two inputs. In fact, human listeners would not start over from the beginning of a sentence as it unfolds word-by-word, but update it continually as each word is heard and use whatever information currently available to build a coherent interpretation (*McRae and Matsuki, 2013*). This difference, however, does not impede our objective of extracting contextualized structural representations at critical points within a sentence. In the case of BERT, the representation of each word is continuously updated in a bidirectional manner as a new word is added. This process accounts for the constraints imposed by all the words of the input and their interactions, forming a coherent interpretation. Nonetheless, for other hypotheses in speech comprehension, such as parallel parsing and how various grammatically correct sentence structures are maintained and compete in the brain, DLMs with recurrent memory might be more suitable. Such models can better stimulate the continuous, dynamic updating of interpretations that characterizes human sentence processing.

In summary, recent developments in DLMs have shown great potential in capturing the dynamic interplay between syntax, semantics, and broader world knowledge that is essential for successful language comprehension. The empirical evidence from this study supports the notion that DLMs, when utilized as potential models of brain computation and integrated with advanced neuroimaging techniques within a well-defined framework, can offer significant insights into human cognition (*Kriegeskorte and Douglas, 2018*; *Doerig et al., 2023*). Future DLMs, especially those with more human-like model architecture (*McClelland et al., 2020*) and subjected to rigorous evaluation (*Binz and Schulz, 2023*), hold the potential to shed light on the neural implementation of various incremental processes that support the rapid transition from sound to meaning in the brain.

## Materials and methods
### Participants
Seventeen right-handed native British English speakers participated in this study and provided written consent. This sample size was determined according to previous MEG studies on speech comprehension (*Choi et al., 2021*; *Lyu et al., 2019*; *Klimovich-Gray et al., 2019*; *Kocagoncu et al., 2017*). One participant was excluded from subsequent analysis due to sleepiness during EMEG scanning, the other 16 participants were included in the following analyses (aged between 19 and 38 y, 26.5 y on average; seven females). All participants had normal hearing, and none had any pre-existing neurological condition or mental health issues. This study was approved by the Cambridge Psychology Research Ethics Committee (reference number PRE2019.051).

### Stimuli
We constructed 60 sets of six spoken sentences (360 in total) with varying sentential structures. As shown in the example sentence set (see *Supplementary file 3*), unambiguous (UNA), high transitivity (HiTrans), and low transitivity (LoTrans) sentences contain a long-distance dependency between the subject noun and the main verb introduced by a full or reduced relative clause inserted in between. Whereas there is no long-distance dependency in the sentences of passive (PAS) and two direct object (DO1 and DO2) conditions.

Unlike the first verb (Verb1) in the UNA sentences which is unambiguously interpreted as the head of a relative clause, Verb1 in both HiTrans and LoTrans sentences can also be considered alternatively as the 'main verb' before the actual main verb (e.g. *was covered* in the example set in *Supplementary file 3*) was heard. By varying the nature of Verb1 in the reduced relative clause (e.g. *found/walked*),

we manipulated the preference for the two plausible structural interpretations in HiTrans and LoTrans sentences before the appearance of the actual main verb (i.e. a passive interpretation where Verb1 is the head of a relative clause – the subject noun undergoes the action specified by Verb1; an active interpretation where Verb1 is the main verb – the subject noun performs the action specified by Verb1).

Specifically, in the LoTrans sentences, Verb1 was selected to be optionally transitive according to CELEX (*Baayen, 1993*) and Google n-gram corpus (books.google.com/ngrams), meaning that it can either take a direct object or not. Thus, a listener could be initially 'garden-pathed' into the alternative active interpretation where the Verb1 is considered as the main verb when a following prepositional phrase fits its intransitive use. Whereas Verb1 in HiTrans sentences was selected to have a higher preference for taking a direct object than Verb1 in LoTrans sentences [subcategorization frame (SCF) probability for direct object according to VALEX (*Korhonen et al., 2006*): HiTrans 0.71 ± 0.16, LoTrans 0.44 ± 0.19, two-tailed two-sample *t*-test, $t_{(117)} = 8.45$, p=$9.3 \times 10^{-14}$]. Therefore, Verb1 in HiTrans sentences was more likely to be recognized as the head of a reduced relative clause given the appearance of a prepositional phrase (e.g. *in the park*) rather than a highly expected direct object.

In all the six types of sentences, the subject noun phrase comprised a single-word noun and a preceding determine '*The*'. In each sentence set, sentences of the first four types had the same subject noun, while DO1 and DO2 sentences shared a different subject noun. The reduced relative clause in HiTrans and LoTrans sentences consisted of a head verb, that is, Verb1 (e.g. *found/walked*) which was followed by a three-word prepositional phrase (e.g. *in the park*). Note that, in 15 out of the 60 sentence sets, the actual main verb in UNA, RR, and GP conditions was preceded by an auxiliary verb (e.g. '*was covered*' in the example set in *Supplementary file 3*). In the following analyses, we defined the first word after the prepositional phrase as the main verb since its appearance is sufficient to resolve the intended passive interpretation where Verb1 is a passive verb (i.e. the head of a reduced relative clause).

Note that, although UNA, PAS, DO1, and DO2 conditions were not included in subsequent analyses, they added variety to the types of syntactic construction of the stimuli and ecological validity of the experiment, which also prevented potential adaption to a particular sentence structure.

## Procedure

The experimental stimuli (360 spoken sentences recorded by a female native British English speaker with a neutral intonation throughout) were equally divided into four blocks with 90 experimental trials in each. To maintain participants' attention while they were listening to the stimuli, the experimental trials in each block were interspersed with nine additional trials consisting of questions related to the contents in the preceding sentence. These questions were presented in written form on the screen, and a 'yes' or 'no' response was required by button pressing. Each of these question trials was followed by a filler trial (including a normal spoken sentence outside the experimental stimuli) to ensure that no residual task effects would be picked up in the next experimental trial. Each block started with two filler trials. All question trials and filler trials (20 in each block) were excluded from the following analyses. The order of blocks and the order of trials within each block were pseudorandomized across participants.

Each experimental trial began with a fixation cross presented at the centre of the screen with a random period ranging from 750 ms to 1250 ms (1000 ms on average) before the onset of the spoken sentence. Participants were asked to look at the fixation cross and avoid eye movement or blinking while listening to the spoken sentences. There was a 1000 ms silence from the end of each sentence followed by a 'blink cue' that lasted for 1400 ms during which participants could blink. E-Prime Studio version 2 (Psychology Software Tools Inc, PA) was used to present stimuli and record participants' responses.

Auditory stimuli were delivered binaurally through MEG-compatible ER3A insert earphones (Etymotic Research Inc, IL). There was a 26 ms ± 2 ms delay in sound delivery due to the transmission of auditory signal from the stimulus computer to the earphones. This sound delivery delay was corrected in the following analyses. To ensure that participants were able to hear the stimuli through both earphones, a short hearing test was conducted before the main experiment.

## Sentence continuation pre-tests

To obtain human incremental interpretations for each of the HiTrans and LoTrans sentences (120 in total), we conducted two continuation pre-tests which involved two different groups of native British English speakers (30 participants in the first pre-test, 18 participants in the second, aged between 18 and 34 y) who did not participate in the main experiment. Specifically, participants wore headphones and were seated in front of a computer. They listened to a fragment of one of the HiTrans/LoTrans sentences starting from its onset and continuing until a certain position in the sentence, and then they were asked to complete this sentence by producing a meaningful continuation. The sentence fragment was binaurally presented up to the Verb1 (e.g. *The dog found …*) in the first pre-test and was presented up to the end of the prepositional phrase in the second pre-test (e.g. *The dog found in the park …*). In the second pre-test, participants were allowed to provide a full stop as a continuation if they thought that what they had heard was a complete sentence.

Based on the continuations obtained in the first pre-test, we calculated the probability of direct object or prepositional phrase in the continuations immediately after the Verb1, that is, DO probability and PP probability. This provided contextualized measures of the transitive or the intransitive use of Verb1 given the preceding subject noun phrase. Specifically, we defined Verb1 transitivity as DO probability/(1 - DO probability) and defined Verb1 intransitivity as (1 - DO probability)/DO probability. Given the continuations collected in the second pre-test, we calculated the probability of a main verb in the continuations immediately after the prepositional phrase, that is, MV probability, which directly reflected a listener's structural interpretation by the end of the prepositional phrase. The absence of a main verb in the continuation after the prepositional phrase indicated that the active interpretation was taken and the Verb1 was considered as the 'main verb', whereas the appearance of a main verb in the continuation indicated that the passive interpretation was taken (i.e. Verb1 was interpreted as a passive verb), and thus a main verb was needed to complete the sentence.

## EMEG and MRI acquisition

Participants were seated in a magnetically shielded room (IMEDCO GmbH, Switzerland) with their head placed in the helmet of the MEG scanner. MEG data were collected using a Neuromag Vector View system (Elekta, Helsinki, Finland) with 102 magnetometers and 204 planar gradiometers at 1 kHz sampling rate. Simultaneous EEG was recorded at 1 kHz sampling rate from 70 Ag–AgCl electrodes within an elastic cap (ESACYCAP GmbH, Herrsching-Breitbrunn, Germany). Vertical and horizontal eye movements were recorded by two EOG electrodes attached below and lateral to the left eye, and cardiac signals were recorded by two ECG electrodes attached separately to the right shoulder blade and left torso. Five head position indicator (HPI) coils were used to monitor head motion. A 3D digitizer was used to record the position of EEG electrodes, HPI coils and head points on participants' scalp relative to the three anatomical fiducials (i.e. nasion and bilateral preauricular points). To source localize EMEG data, T1-weighted MPRAGE structural MRI image with 1 mm isotropic resolution was acquired using a Siemens Prisma 3T scanner (Siemens, Erlangen, Germany). All EMEG and MRI data were collected at the MRC Cognition and Brain Sciences Unit, University of Cambridge.

## EMEG preprocessing and source localization

Maxfilter (Elekta) was applied to raw MEG data for bad channel removal and head-motion compensation. Signals outside the brain were removed using the temporal extension of signal-space separation (*Taulu and Simola, 2006*). EMEG data were low-pass filtered at 40 Hz and high-pass filtered at 0.5 Hz with a fifth-order bidirectional Butterworth filter using SPM12 (Wellcome Trust Centre for Neuroimaging, UCL). Independent component analysis (ICA) was conducted using EEGLAB (SCCN, UCSD), components related to blink, eye-movement, and physiological noises were removed according to the correlation with EOG, ECG signals, and further visual inspection. The preprocessed EMEG data were then downsampled to 200 Hz. Three epochs were extracted from the continuous EMEG recordings of each HiTrans or LoTrans sentence with auditory delivery delay corrected – V1 epoch was aligned to the onset of the Verb1, PP1 epoch was aligned to the onset of the preposition, and MV epoch was aligned to the onset of the main verb. All the three epochs were 600 ms in length. For all three epochs, baseline correction was performed using the signal from a silent period (i.e. –200 ms to 0 ms relative to sentence onset). Finally, automatic artefact rejection was conducted to exclude trials with signals that exceeded predefined amplitude thresholds (60 ft/mm for gradiometers, 3000 ft for

magnetometers, and 200 uV for EEG electrodes). The uniqueness point of V1/PP1/MV was defined as the earliest point in time when this word can be fully recognized after removing all of its phonological competitors. We first identified the phoneme by which this word can be uniquely recognized according to CELEX (*Baayen, 1993*). Then, we manually labelled the offset of this phoneme in the auditory file of the spoken sentence.

EMEG data source localization was performed using SPM12. Source space was modelled by a cortical mesh consisting of 8196 vertices. The sensor positions were co-registered to individual T1-weighted structural image by aligning fiducials and the digitized head shape to the outer scalp mesh. MEG forward model was constructed using the single-shell model (*Sarvas, 1987*), while EEG forward model was built using the boundary element model (*Mosher et al., 1999*). Inversion of EMEG data was conducted for V1, PP1, and MV epochs separately using the least-squares minimum norm method (*Hämäläinen and Ilmoniemi, 1994*) and an empirical Bayesian MEG and EEG data fusion scheme implemented in SPM12 (*Henson et al., 2009*).

## Incremental structural representations of BERT

To obtain incremental structural representations of BERT, we adopted a structural probing approach (*Hewitt and Manning, 2019*) to quantify a sentence's structure by estimating each word's parse depth in the corresponding dependency parse tree based on the contextualized word embeddings from the hidden states of BERT, which explicitly considers the specific contents of the words in the input. Specifically, a structural probing model was trained to find an optimal linear transformation to be applied to the BERT contextualized embeddings of words in the input sentence, so that the squared L2 norm of the transformed word embeddings provided the best estimate for each word's parse depth of in the dependency parse tree of this sentence.

We followed the procedure described in a previous study (*Hewitt and Manning, 2019*) and trained a structural probing model for each BERT layer with the annotated corpus from Penn Treebank (*Marcus et al., 1993*). Contextualized word embeddings were extracted from each of the 24 layers in a pre-trained version of BERT (BERT-large-cased) using HuggingFace (*Wolf, 2019*). For each BERT layer, the training process was repeated 10 times with different random initializations, and the averaged BERT parse depth was used in the following analyses. The performance of structural probing models trained by different BERT layers was evaluated by root accuracy. Root accuracy is defined as the percentage of the sentences in which the smallest parse depth is assigned to the main verb (i.e. the root of the dependency parse tree has a parse depth of 0) when the whole sentence is input to the model.

Each HiTrans or LoTrans sentence was input word-by-word to the trained BERT structural probing models, which resulted in a vector consisting of the parse depth of each word in the incremental input (e.g. a3D BERT parse depth vector for the input "*The dog found …*"). Taking advantage of the bidirectional attention mechanism of BERT, the parse depth of each preceding word was constantly updated as the input unfolded word-by-word, capturing the incrementality of speech comprehension. Besides, we defined interpretive mismatch as the cosine distance between an incremental BERT parse depth vector and the corresponding incremental context-free parse depth vector for the passive or the active interpretation. The smaller the interpretive mismatch with one particular interpretation, the higher the preference for this interpretation given BERT structural representations.

To determine the contribution of the words at different positions in a sentence to the incremental BERT parse depth vectors, we shuffled the parse depths of the words at a particular position across sentences at a time and kept the parse depths of the other words unchanged. Then we calculated the Spearman distance (i.e. 1-Spearman's rho) between the original BERT parse depth vector and the shuffled counterpart. The higher this distance, the more important the words at this position are to the BERT parse depth vectors (see *Figure 3—figure supplement 3*).

## Corpus-based measures of multifaceted constraints and their interpretative coherence

We quantified subject noun thematic role properties and Verb1 transitivity preference based on a concatenated corpus (3.4 billion tokens, 162.1 million sentences) consisting of the British National Corpus, the Wikipedia dump (by October 2020) and ukWaC (*Baroni et al., 2009*). A dependency parser (*Mrini et al., 2020*) was first applied to each sentence in the concatenated corpus to specify the subject noun and the verb(s) related to it. For the subject noun in each sentence, we used a semantic

role labelling model (*Li et al., 2020*) to obtain its thematic role. For simplicity, we only considered the thematic role of an agent or a patient (*Dowty, 1991*).

For each subject noun in HiTrans and LoTrans sentences, we counted separately how many times it took the thematic role of an agent or a patient in the concatenated corpus. Then we defined its agenthood as the ratio of the number of its appearances as an agent to that of its appearances as a patient, and vice versa for its patienthood. We also counted separately how many times the Verb1 in each HiTrans or LoTrans sentence took a direct object or alternative SCFs in the corpus. Each Verb1's transitivity was defined as the ratio of the frequency it took a direct object to that it took alternative SCFs, and vice versa for its intransitivity. By doing so, we obtained subject noun agenthood and patienthood, Verb1 transitivity and intransitivity. Note that the Verb1 (in)transitivity estimated from the human continuations in the pre-test is context-dependent given the specific preceding subject noun, while the Verb1 (in)transitivity estimated from the concatenated corpus is context-independent in the sense that it accounted for every appearance of this verb without being biased to a specific context.

We further derived passive/active index capturing the interpretative coherence between the subject noun and Verb1 as they affected passive and active interpretations separately. The passive index was obtained by multiplying subject noun patienthood with Verb1 transitivity, given that both high subject noun patienthood and high Verb1 transitivity coherently prefer a passive interpretation as the prepositional phrase is heard (e.g. *The **dog found** in the park* …). In contrast, the active index was obtained by multiplying subject noun agenthood by Verb1 intransitivity, capturing the preference for an active interpretation (e.g. *The **king walked** in the garden* …). Besides, we also calculated a contextualized version of passive/active index by using Verb1 (in)transitivity derived from human continuation pre-tests instead of that derived from the concatenated corpus. In addition, we derived a non-directional index by applying logarithmic transformation to the ratio measures of lexical constraints (i.e. subject noun agenthood or patienthood, Verb1 intransitivity or transitivity) before multiplying them. This manipulation removed the directionality of the passive or the active index, so that the non-directional index only indicates the interpretative coherence between the subject noun and Verb1 regardless of which interpretation is considered (see illustrations of directionality in *Figure 8—figure supplement 3*).

## Spatiotemporal searchlight representational similarity analysis (ssRSA)

ssRSA was conducted to compare the (dis)similarity structure of BERT structural measures or the multifaceted probabilistic constraints with the (dis)similarity structure of observed spatiotemporal patterns of listeners' brain activity. We used a spatiotemporal searchlight with a 10 mm spatial radius and 30 ms temporal radius (i.e. a 60 ms sliding time window) which was mapped across the whole brain in the source-localized EMEG.

For brain activity, we constructed data representational dissimilarity matrix (RDM) by vectorizing the source-localized EMEG data within each spatiotemporal searchlight for all the trials (i.e. 60 HiTrans sentences and 60 LoTrans sentences) and calculated the pair-wise Pearson's correlation distance (i.e. 1 - Pearson's r) among them, which resulted in a 120 × 120 data RDM. Multivariate normalization was applied to improve the reliability of distance measures and reduce the task-irrelevant heteroscedastic structure across trials and vertices (*Guggenmos et al., 2018*). Model RDMs of the same size (i.e. 120 × 120) were constructed by calculating either the absolute pair-wise difference for a scalar measure (e.g. SN agenthood/patienthood, passive/active index, BERT Verb1 parse depth) or the cosine distance among the incremental BERT parse depth vectors of the 120 sentences (see *Supplementary file 1* for a summary of all the model RDMs). We used ratio measures to represent subject noun agenthood/patienthood, Verb1 transitivity/intransitivity, and passive/active index because they provided the directionality needed to differentiate the two opposite aspects of the same lexical constraint in the model RDMs (see illustrations in *Figure 8—figure supplement 3*) and made it possible to test them separately in the brain.

Each model RDM was compared against the data RDM of a searchlight centred at each vertex and time point using Spearman's rank correlation, which resulted in a time series of model fit (i.e. rank correlation coefficient rho) for each vertex. For each time point, a one-tailed one-sample *t*-test was conducted at each vertex with the fits of all participants for this model RDM to test whether the mean model fit is significantly above zero. Cluster permutation tests were performed for multiple comparison correction with 5000 nonparametric permutations, vertex-wise p<0.01 and cluster-wise p<0.05.

## GCA based on ssRSA model fits

GCA was conducted to investigate the relationship between multifaceted constraints and BERT structural measures in terms of their effects in the brain, that is, ssRSA model fits of the corresponding model RDMs. For a given model RDM, each participant's non-thresholded whole-brain model fit time series were normalized and concatenated across participants, which resulted in a vertex by time point matrix. Non-negative matrix factorization (NMF) was applied to this concatenated model fit matrix with negative model fits zeroed. NMF was repeated 20 times with random starting values using a multiplicative update algorithm in MATLAB, results with the least root mean square residual (RMS) was used in the following analyses. The optimal number of NMF factors was determined by searching for the one with the least RMS in a wide range of factor numbers (from 2 to 50). With the optimal number of factors, the resulted time series of NMF factors from all participants were reshaped into a factor by time point by participant matrix which was used as the input of multivariate GCA implemented by the Multivariate GCA toolbox (*Barnett and Seth, 2014*). Multivariate GCA was conducted using two sets of NMF factors derived from the fits of two model RDMs with Akaike information criterion adopted for GCA model order estimation. GC significance was determined by a permutation test in which 1000 surrogate data sets were created by randomly rearranging short time windows (of length model order) from the original factor time course. p-Values <0.005 were refined via a tail approximation from the Generalised Pareto Distribution using PALM (*Winkler et al., 2016*). Multiple comparisons were controlled with false discovery rate (FDR) alpha < 0.05 for both between- and within-model RDM GC connections.

## Acknowledgements

This research was funded by the European Research Council Advanced Investigator Grant to LKT under the European Community's Horizon 2020 Research and Innovation Programme (2014–2022 ERC Grant Agreement 669820). BL was supported by the Ministry of Science and Technology of China and Changping Laboratory. We thank Billi Randall and Barry Devereux for their valuable contributions to early experimental design and to stimulus development; and Hun S Choi, Benedict Vassileiou, John Hewitt, Tao Li, Yi Zhu, Nai Ding, and Giorgio Marinato for helpful discussions.

## Additional information

### Funding

| Funder | Grant reference number | Author |
| --- | --- | --- |
| European Research Council | 669820 | Lorraine K Tyler |
| Ministry of Science and Technology of the People's Republic of China | | Bingjiang Lyu |
| Chanping Laboratory | | Bingjiang Lyu |

The funders had no role in study design, data collection and interpretation, or the decision to submit the work for publication.

### Author contributions

Bingjiang Lyu, Conceptualization, Data curation, Formal analysis, Funding acquisition, Investigation, Methodology, Visualization, Writing – original draft, Writing – review and editing; William D Marslen-Wilson, Conceptualization, Writing – original draft, Methodology; Yuxing Fang, Supervision, Writing – review and editing; Lorraine K Tyler, Conceptualization, Writing – original draft, Data curation, Project administration, Methodology

### Author ORCIDs

Bingjiang Lyu https://orcid.org/0000-0001-8554-5138

### Ethics

All participants in this study provided written consent. This study was approved by the Cambridge Psychology Research Ethics Committee (reference number PRE2019.051).

Reviewer #2 (Public Review): https://doi.org/10.7554/eLife.89311.3.sa1

Reviewer #3 (Public Review): https://doi.org/10.7554/eLife.89311.3.sa2

Author response https://doi.org/10.7554/eLife.89311.3.sa3

## Additional files

### Supplementary files

• MDAR checklist

• Supplementary file 1. Summary of all quantitative measures used to create model RDMs.

• Supplementary file 2. Full anatomical labels of ROI abbreviations in ssRSA results.

• Supplementary file 3. Example stimuli sentence set presented to both BERT and human listeners.

### Data availability

Preprocessed E/MEG data, scripts and relevant materials are available in the following repository https://osf.io/7u8jp/.

The following dataset was generated:

| Author(s) | Year | Dataset title | Dataset URL | Database and Identifier |
|---|---|---|---|---|
| Lyu B, Marslen-Wilson WD, Fang Y, Tyler LK | 2024 | Finding structure during incremental speech comprehension | https://osf.io/7u8jp/ | Open Science Framework, 10.17605/OSF.IO/7U8JP |

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
