## [Editor Report · eLife assessment]

This **valuable** study provides insights into how the brain parses the syntactic structure of a spoken sentence. **Convincing** evidence is provided that distributive cortical networks are engaged for incremental parsing of a sentence, and neural activity recorded by MEG correlates with sentence structure measures extracted by a deep neural network language model, that is, BERT. A contribution of the work is to use a deep neural network model to quantify how the mental representation of syntactic structure updates as a sentence unfolds in time.

---

## [Referee Report · Reviewer #2 (Public Review)]

This article is focused on investigating incremental speech processing, as it pertains to building higher order syntactic structure. This is an important question because speech processing in general is lesser studied as compared to reading, and syntactic processes are lesser studied than lower-level sensory processes. The authors claim to shed light on the neural processes that build structured linguistic interpretations. The authors apply modern analysis techniques, and use state-of-the-art large language models in order to facilitate this investigation. They apply this to a cleverly designed experimental paradigm of EMEG data, and compare neural responses of human participants to the activation profiles in different layers of the BERT language model.

Comments on revised version:

Similar to my original review, I find the paper hard to follow, and it is not clear to me that the use of the LLM is adding much to the findings. Using complex language models without substantial motivation dampens my enthusiasm significantly. This concern has not been alleviated since my original review.

---

## [Referee Report · Reviewer #3 (Public Review)]

Syntactic parsing is a highly dynamic process: When an incoming word is inconsistent with the presumed syntactic structure, the brain has to reanalyze the sentence and construct an alternative syntactic structure. Since syntactic parsing is a hidden process, it is challenging to describe the syntactic structure a listener internally constructs at each time moment. Here, the authors overcome this problem by (1) asking listeners to complete a sentence at some break point to probe the syntactic structure mentally constructed at the break point, and (2) using a DNN model to extract the most likely structure a listener may extract at a time moment.

After obtaining incremental syntactic features using a DNN model, i.e., BERT, the authors analyze how these syntactic features are represented in the brain using MEG. The advantage of the approach is that BERT can potentially integrate syntactic and semantic knowledge and is a computational model, instead of a static theoretical construct, that may more precisely reflect incremental sentence processing in the human brain. The results indeed confirm the similarity between MEG activity and measures from the BERT model.

---

## [Author Response]

The following is the authors’ response to the original reviews.

**eLife assessment**
This study provides valuable insights into how the brain parses the syntactic structure of a spoken sentence. A unique contribution of the work is to use a large language model to quantify how the mental representation of syntactic structure updates as a sentence unfolds in time. Solid evidence is provided that distributive cortical networks are engaged for incremental parsing of a sentence, although the contribution could be further strengthened if the authors would further highlight the main results and clarify the benefit of using a large language model.

We thank the editors for the overall positive assessment. We have revised our manuscript to further emphasize our main findings and highlight the advantages of using a large language model (LLM) over traditional behavioural and corpus-based data.

This study aims to investigate the neural dynamics underlying the incremental construction of structured interpretation during speech comprehension. While syntactic cues play an important role, they alone do not define the essence of this parsing process. Instead, this incremental process is jointly determined by the interplay of syntax, semantics, and non-linguistic world knowledge, evoked by the specific words heard sequentially by listeners. To better capture these multifaceted constraints, we derived structural measures from BERT, which dynamically represent the evolving structured interpretation as a sentence unfolds word-by-word.

Typically, the syntactic structure of a sentence can be represented by a context-free parse tree, such as a dependency parse tree or a constituency-based parse tree, which abstracts away from specific content, assigning a discrete parse depth to each word regardless of its semantics. However, this context-free parse tree merely represents the result rather than the process of sentence parsing and does not elucidate how a coherent structured interpretation is concurrently determined by multifaceted constraints. In contrast, BERT parse depth, trained to approach the context-free discrete dependency parse depth, is a continuous variable. Crucially, its deviation from the corresponding discrete parse depth indicates the preference for the syntactic structure represented by this context-free parse. As BERT processes a sentence delivered word-by-word, the dynamic change of BERT parse depth reflects the incremental nature of online speech comprehension.

Our results reveal a behavioural alignment between BERT parse depth and human interpretative preference for the same set of sentences. In other words, BERT parse depth could represent a probabilistic interpretation of a sentence’s structure based on its specific contents, making it possible to quantify the preference for each grammatically correct syntactic structure during incremental speech comprehension. Furthermore, both BERT and human interpretations show correlations with linguistic knowledge, such as verb transitivity, and non-linguistic knowledge, like subject noun thematic role preference. Both types of knowledge are essential for achieving a coherent interpretation, in accordance with the “constraint-based hypothesis” of sentence processing.

Motivated by the observed behavioural alignment between BERT and human listeners, we further investigated BERT structural measures in source-localized EEG/MEG using representational similarity analyses (RSA). This approach revealed the neural dynamics underlying incremental speech comprehension on millisecond scales. Our main findings include: (1) a shift from bi-hemispheric lateral frontal-temporal regions to left-lateralized regions in representing the current structured interpretation as a sentence unfolds, (2) a pattern of sequential activations in the left lateral temporal regions, updating the structured interpretation as syntactic ambiguity is resolved, and (3) the influence of lexical interpretative coherence activated in the right hemisphere over the resolved sentence structure represented in the left hemisphere.

From our perspective, the advantages of using a LLM (or deep language model) like BERT are twofold. Conceptually, BERT structural measures offer a deep contextualized structural representation for any given sentence by integrating the multifaceted constraints unique to the specific contents described by the words within that sentence. Modelling this process on a word-by-word basis is challenging to achieve with behavioural or corpus-based metrics. Empirically, as demonstrated in our responses to the reviewers below, BERT measures show better performance compared to behavioural and corpus-based metrics in aligning with listeners’ neural activity. Moreover, when it comes to integrating multiple sources of constraints for achieving a coherent interpretation, BERT measures also show a better fit with the behavioural data of human listeners than corpus-based metrics.

Taken together, we propose that LLMs, akin to other artificial neural networks (ANNs), can be considered as computational models for formulating and testing specific neuroscientific hypotheses, such as the “constraint-based hypothesis” of sentence processing in this study. However, we by no means overlook the importance of corpus-based and behavioural metrics. These metrics play a crucial role in interpreting and assessing whether and how ANNs stimulate human cognitive processes, a fundamental step in employing ANNs to gain new insights into the neural mechanisms of human cognition.

**Public Reviews:**

**Reviewer #1 (Public Review):**
In this study, the authors investigate where and when brain activity is modulated by incoming linguistic cues during sentence comprehension. Sentence stimuli were designed such that incoming words had varying degrees of constraint on the sentence's structural interpretation as participants listened to them unfolding, i.e. due to varying degrees of verb transitivity and the noun's likelihood of assuming a specific thematic role. Word-by-word "online" structural interpretations for each sentence were extracted from a deep neural network model trained to reproduce language statistics. The authors relate the various metrics of word-by-word predicted sentence structure to brain data through a standard RSA approach at three distinct points of time throughout sentence presentation. The data provide convincing evidence that brain activity reflects preceding linguistic constraints as well as integration difficulty immediately after word onset of disambiguating material.

We thank Reviewer #1 (hereinafter referred to as R1) for their recognition of the objectives of our study and the analytical approaches we have employed in this study.

The authors confirm that their sentence stimuli vary in degree of constraint on sentence structure through independent behavioral data from a sentence continuation task. They also show a compelling correlation of these behavioral data with the online structure metric extracted from the deep neural network, which seems to pick up on the variation in constraints. In the introduction, the authors argue for the potential benefits of using deep neural networkderived metrics given that it has "historically been challenging to model the dynamic interplay between various types of linguistic and nonlinguistic information". Similarly, they later conclude that "future DLMs (...) may provide new insights into the neural implementation of the various incremental processing operations(...)".

We appreciate R1’s positive comments on the design, quantitative modelling and behavioural validation of the sentence stimuli used in this experiment.

By incorporating structural probing of a deep neural network, a technique developed in the field of natural language processing, into the analysis pipeline for investigating brain data, the authors indeed take an important step towards establishing advanced machine learning techniques for researching the neurobiology of language. However, given the popularity of deep neural networks, an argument for their utility should be carefully evidenced.

We fully concur with R1 regarding the need for cautious evaluation and interpretation of deep neural networks’ utility. In fact, this perspective underpinned our decision to conduct extensive correlation analyses using both behavioural and corpus-based metrics to make sense of BERT metrics. These analyses were essential to interpret and validate BERT metrics before employing them to investigate listeners’ neural activity during speech comprehension. We do not in any way undermine the importance of behavioural or corpus-based data in studying language processing in the brain. On the contrary, as evidenced by our findings, these traditional metrics are instrumental in interpreting and guiding the use of metrics derived from LLMs.

However, the data presented here don't directly test how large the benefit provided by this tool really is. In fact, the authors show compelling correlations of the neural network-derived metrics with both the behavioral cloze-test data as well as several (corpus-)derived metrics. While this is a convincing illustration of how deep language models can be made more interpretable, it is in itself not novel. The correlation with behavioral data and corpus statistics also raises the question of what is the additional benefit of the computational model? Is it simply saving us the step of not having to collect the behavioral data, not having to compute the corpus statistics or does the model potentially uncover a more nuanced representation of the online comprehension process? This remains unclear because we are lacking a direct comparison of how much variance in the neural data is explained by the neural network-derived metrics beyond those other metrics (for example the main verb probability or the corpusderived "active index" following the prepositional phrase).

From our perspective, a primary advantage of using the neural network-derived metrics (or LLMs as computational models of language processing), compared to traditional behavioural and corpus-based metrics, lies in their ability to offer more nuanced, contextualized representations of natural language inputs. There seems no effective way of computationally capturing the distributed and multifaceted constraints within specific contexts until the current generation of LLMs came along. While it is feasible to quantify lexical properties or contextual effects based on the usage of specific words via corpora or behavioural tests, this method appears less effective in modelling the composition of meanings across more words on the sentence level. More critically, it struggles with capturing how various lexical constraints collectively yield a coherent structured interpretation.

Accumulating evidence suggests that models designed for context prediction or next-word prediction, such as word2vec and LLMs, outperform classic count-based distributional semantic models (Baroni et al. 2014) in aligning with neural activity during language comprehension (Schrimpf et al. 2021; Caucheteux and King 2022). Relevant to this, we have conducted additional analyses to directly assess the additional variance of neural data explained by BERT metrics, over and above what traditional metrics account for. Specifically, using RSA, we re-tested model RDMs based on BERT metrics while controlling for the contribution from traditional metrics (via partial correlation).

During the first verb (V1) epoch, we tested model RDMs of V1 transitivity based on data from either the behavioural pre-test (i.e., continuations following V1) or massive corpora. Contrasting sharply with the significant model fits observed for BERT V1 parse depth in bilateral frontal and temporal regions, the two metrics of V1 transitivity did not exhibit any significant effects (see Author response image 1).

Author response image 1

RSA model fits of BERT structural metrics and behavioural/corpus-based metrics in the V1 epoch. (upper) Model fits of BERT V1 parse depth (relevant to Appendix 1-figure 10A); (middle) Model fits of the V1 transitivity based on the continuation pre-rest conducted at the end of V1 (e.g., completing “The dog found …”); (bottom) Model fits of the V1 transitivity based on the corpus data (as described in Methods). Note that verb transitivity is quantified as the proportion of its transitive uses (i.e., followed by a direct object) relative to its intransitive uses.

In the PP1 epoch, which was aligned to the onset of the preposition in the prepositional phrase (PP), we tested the probability of a PP continuation following V1 (e.g., the probability of a PP after “The dog found…”). While no significant results were found for PP probability, we have plotted the uncorrected results for PP probability (Author response image 2). These model fits have very limited overlap with those of BERT parse depth vector (up to PP1) in the left inferior frontal gyrus (approximately at 360 ms) and the left temporal regions (around 600 ms). It is noteworthy that the model fits of the BERT parse depth vector (up to PP1) remained largely unchanged even when PP probability was controlled for, indicating that the variance explained by BERT metrics cannot be effectively accounted for by the PP probability obtained from the human continuation pre-test.

Author response image 2

Comparison between the RSA model fits of BERT structural metrics and behavioural / corpusbased metrics in the PP1 epoch. (upper) Model fits of BERT parse depth vector up to PP1 (relevant to Figure 6B in the main text); (middle) Model fits of the probability of a PP continuation in the prerest conducted at the end of the first verb; (bottom) Model fits of BERT parse depth vector up to PP1 after partialling out the variance explained by PP probability.

**Author response image 2. sa3fig2:** 

Finally, in the main verb (MV) epoch, we tested the model RDM based on the probability of a MV continuation following the PP (e.g., the probability after “The dog found in the park…”). When compared with the BERT parse depth vector (up to MV), we observed a similar effect in the left dorsal frontal regions (see Author response image 3). However, this effect did not survive after the whole-brain multiple comparison correction. Subsequent partial correlation analyses revealed that the MV probability accounted for only a small portion of the variance in neural data explained by the BERT metric, primarily the effect observed in the left dorsal frontal regions around 380 ms post MV onset. Meanwhile, the majority of the model fits of the BERT parse depth vector remained largely unchanged after controlling for the MV probability.

Note that the probability of a PP/MV continuation reflect participants’ predictions based on speech input preceding the preposition (e.g., “The dog found…”) or the main verb (e.g., “The dog found in the park…”), respectively. In contrast, BERT parse depth vector is designed to represent the structure of the (partial) sentence in the speech already delivered to listeners, rather than to predict a continuation after it. Therefore, in the PP1 and MV epochs, we separately tested BERT parse depth vectors that included the preposition (e.g., “The dog found in…”) and the main verb (e.g., “The dog found in the park was…”) to accurately capture the sentence structure at these specific points in a sentence. Despite the differences in the nature of information captured by these two types of metrics, the behavioural metrics themselves did not exhibit significant model fits when tested against listeners’ neural activity.

Author response image 3

Comparison between the RSA model fits of BERT structural metrics and behavioural / corpusbased metrics in the MV epoch. (upper) Model fits of BERT parse depth vector up to MV (relevant to Figure 6C in the main text); (middle) Model fits of the probability of a MV continuation in the pre-rest conducted at the end of the prepositional phrase (e.g., “The dog found in the park …”); (bottom) Model fits of BERT parse depth vector up to MV after partialling out the variance explained by MV probabilit.

**Author response image 3. sa3fig3:** 

Regarding the corpus-derived interpretative preference, we observed that neither the Active index nor the Passive index showed significant effects in the PP1 epoch. In the MV epoch, while significant model fits of the passive index were observed, which temporally overlapped with the BERT parse depth vector (up to MV) after the recognition point of the MV, the effects of these two model RDMs emerged in different hemispheres, as illustrated in Figures 6C and 8D in the main text. Consequently, we opted not to pursue further partial correlation analysis with the corpus-derived interpretative preference. Besides, as shown in Figure 8A, 8B and 8C, subject noun thematic role preference and non-directional index exhibit significant model fits in the PP1 or the MV epoch. Interesting, these effects lead corresponding effects of BERT metrics in the same epoch (see Figure 6B and 6C), suggesting that the overall structured interpretation emerges after the evaluation and integration of multifaceted lexical constraints.

In summary, our findings indicate that, in comparison to corpus-derived or behavioural metrics, BERT structural metrics are more effective in explaining neural data, in terms of modelling both the unfolding sentence input (i.e., incremental BERT parse vector) and individual words (i.e., V1) within specific sentential contexts. This advantage of BERT metrics might be due to the hypothesized capacity of LLMs to capture more contextually rich representations. Such representations effectively integrate the diverse constraints present in a given sentence, thereby outperforming corpus-based metrics or behavioural metrics in this respect. Concurrently, it is important to recognize the significant role of corpus-based / behavioral metrics as explanatory variables. They are instrumental not only in interpreting BERT metrics but also in understanding their fit to listeners’ neural activity (by examining the temporal sequence and spatial distribution of model fits of these two types of metrics). Such an integrative approach allows for a more comprehensive understanding of the complex neural processes underpinning speech comprehension.

With regards to the neural data, the authors show convincing evidence for early modulations of brain activity by linguistic constraints on sentence structure and importantly early modulation by the coherence between multiple constraints to be integrated. Those modulations can be observed across bilateral frontal and temporal areas as well as parts of the default mode network. The methods used are clear and rigorous and allow for a detailed exploration of how multiple linguistic cues are neurally encoded and dynamically shape the final representation of a sentence in the brain. However, at times the consequences of the RSA results remain somewhat vague with regard to the motivation behind different metrics and how they differ from each other. Therefore, some results seem surprising and warrant further discussion, for example: Why does the neural network-derived parse depth metric fit neural data before the V1 uniqueness point if the sentence pairs begin with the same noun phrase? This suggests that the lexical information preceding V1, is driving the results. However, given the additional results, we can already exclude an influence of subject likelihood for a specific thematic role as this did not model the neural data in the V1 epoch to a significant degree.

As pointed out by R1, model fits of BERT parse depth vector (up to V1) and its mismatch for the active interpretation were observed before the V1 uniqueness point (Figures 6A and 6D). These early effects could be attributed to the inclusion of different subject nouns in the BERT parse depth vectors. In our MEG data analyses, RSA was performed using all LoTrans and HiTrans sentences. Each of the 60 sentence sets contained one LoTrans sentence and one HiTrans sentence, which resulted in a 120 x 120 neural data RDM for each searchlight ROI across the brain within each sliding time window. Although LoTrans and HiTrans sentences within the same sentence set shared the same subject noun, subject nouns varied across sentence sets. This variation was expected to be reflected in both the model RDM of BERT metrics and the data RDM, a point further clarified in the revised manuscript.

In contrast, when employing a model RDM constructed solely from the BERT V1 parse depth, we observed model fits peaking precisely at the uniqueness point of V1 (see Appendix 1figure 10). It is important to note that BERT V1 parse depth is a contextualized metric influenced by the preceding subject noun, which could account for the effects of BERT V1 parse depth observed before the uniqueness point of V1.

Relatedly, In Fig 2C it seems there are systematic differences between HiTrans and LoTrans sentences regarding the parse depth of determiner and subject noun according to the neural network model, while this is not expected according to the context-free parse.

We thank R1 for pointing out this issue. Relevant to Figure 3D (Figure 2C in the original manuscript), we presented the distributions of BERT parse depth for individual words as the sentence unfolds in Appendix 1-figure 2. Our analysis revealed that the parse depth of the subject noun in high transitivity (HiTrans) and low transitivity (LoTrans) sentences did not significantly differ, except for the point at which the sentence reached V1 (two-tailed twosample t-test, P = 0.05).

However, we observed a significant difference in the parse depth of the determiner between HiTrans and LoTrans sentences (two-tailed two-sample t-test, P < 0.05 for all results in Appendix 1-figure 2). Additionally, the parse depth of the determiner was found to covary with that of V1 as the input unfolded to different sentence positions (Pearson correlation, P < 0.05 for all plots in Appendix 1-figure 2). This difference, unexpected in terms of the contextfree (dependency) parse used for training the BERT structural probing model, might be indicative of a “leakage” of contextual information during the training of the structural probing model, given the co-variation between the determiner and V1 which was designed to be different in their transitivity in the two types of sentences.

Despite such unexpected differences observed in the BERT parse depths of the determiner, we considered the two sentence types as one group with distributed features (e.g., V1 transitivity) in the RSA, and used the BERT parse depth vector including all words in the sentence input to construct the model RDMs. Moreover, as indicated in Appendix 1-figure 3, compared to the content words, the determiner contributed minimally to the incremental BERT parse depth vector. Consequently, the noted discrepancies in BERT parse depth of the determiner between HiTrans and LoTrans sentences are unlikely to significantly bias our RSA results.

"The degree of this mismatch is proportional to the evidence for or against the two interpretations (...). Besides these two measures based on the entire incremental input, we also focused on Verb1 since the potential structural ambiguity lies in whether Verb1 is interpreted as a passive verb or the main verb." The neural data fits in V1 epoch differ in their temporal profile for the mismatch metrics and the Verb 1 depth respectively. I understand the "degree of mismatch" to be a measure of how strongly the neural network's hidden representations align with the parse depth of an active or passive sentence structure. If this is correct, then it is not clear from the text how far this measure differs from the Verb 1 depth alone, which is also indicating either an active or passive structure.

Within the V1 epoch, we tested three distinct types of model RDMs based on BERT metrics: (1) The BERT parse depth vector, representing the neural network’s hidden representation of the incremental sentence structure including all words up to V1. (2) The mismatch metric for either the Active or Passive interpretation, calculated as the distance between the BERT parse depth vector and the context-free parse depth vector for each interpretation. (3) The BERT parse depth of V1, crucial in representing the preferred structural interpretation of the unfolding sentence given its syntactic role as either a passive verb or the main verb.

While the BERT parse depth vector per se does not directly indicate a preferred interpretation, its mismatch with the context-free parse depth vectors of the two possible interpretations reveals the favoured interpretation, as significant neural fit is only anticipated for the mismatch with the interpretation being considered. The contextualized BERT depth of V1 is also indicative of the preferred structure given the context-free V1 parse depth corresponding to different syntactic roles, however, compared to the interpretative mismatch, it does not fully capture contributions from other words in the input. Consequently, we expected the interpretative mismatch and the BERT V1 depth to yield different results. Indeed, our analysis revealed that, although both metrics extracted from the same BERT layer (i.e., layer 13) demonstrated early RSA fits in the left fronto-temporal regions, the V1 depth showed relatively more prolonged effects with a notable peak occurring precisely at the uniqueness point of V1 (compare Figure 6C and Appendix 1-figure 10). These complementary results underscore the capability of BERT metrics to align with neural responses, in terms of both an incrementally unfolding sentence and a specific word within it.

In previous studies, differences in neural activity related to distinct amounts of open nodes in the parse tree have been interpreted in terms of distinct working memory demands (Nelson et al. pnas 2017, Udden et al tics 2020). It seems that some of the metrics, for example the neural network-derived parse depth or the V1 depth may be similarly interpreted in the light of working memory demands. After all, during V1 epoch, the sentences do not only differ with respect to predicted sentence structure, but also in the amount of open nodes that need to be maintained. In the discussion, however, the authors interpret these results as "neural representations of an unfolding sentence's structure".

We agree with the reviewer that the Active and Passive interpretations differ in terms of the number of open nodes before the actual main verb is heard. Given the syntactic ambiguity in our sentence stimuli (i.e., LoTrans and Hi Trans sentences), it is infeasible to determine the exact number of open nodes in each sentence as it unfolds. Nevertheless, the RSA fits observed in the dorsal lateral frontal regions could be indicative of the varying working memory demands involved in building the structured interpretations across sentences. We have added this perspective in the revised manuscript.

**Reviewer #2 (Public Review):**
This article is focused on investigating incremental speech processing, as it pertains to building higher-order syntactic structure. This is an important question because speech processing in general is lesser studied as compared to reading, and syntactic processes are lesser studied than lower-level sensory processes. The authors claim to shed light on the neural processes that build structured linguistic interpretations. The authors apply modern analysis techniques, and use state-of-the-art large language models in order to facilitate this investigation. They apply this to a cleverly designed experimental paradigm of EMEG data, and compare neural responses of human participants to the activation profiles in different layers of the BERT language model.

We thank Reviewer #2 (hereinafter referred to as R2) for the overall positive remarks on our study.

Strengths:(1) The study aims to investigate an under-explored aspect of language processing, namely syntactic operations during speech processing(2) The study is taking advantage of technological advancements in large language models, while also taking linguistic theory into account in building the hypothesis space(3) The data combine EEG and MEG, which provides a valuable spatio-temporally resolved dataset(4) The use of behavioural validation of high/low transitive was an elegant demonstration of the validity of their stimuli

We thank R2 for recognizing and appreciating the motivation and the methodology employed in this study.

Weaknesses:(1) The manuscript is quite hard to understand, even for someone well-versed in both linguistic theory and LLMs. The questions, design, analysis approach, and conclusions are all quite dense and not easy to follow.

To address this issue, we have made dedicated efforts to clarify the key points in our study. We also added figures to visualize our experimental design and methods (see Figure 1, Figure 3C and Figure 5 in the revised main text). We hope that these revisions have made the manuscript more comprehensible and straightforward for the readers.

(2) The analyses end up seeming overly complicated when the underlying difference between sentence types is a simple categorical distinction between high and low transitivity. I am not sure why tree depth and BERT are being used to evaluate the degree to which a sentence is being processed as active or passive. If this is necessary, it would be helpful for the authors to motivate this more clearly.

Indeed, as pointed by R2, the only difference between LoTrans and HiTrans sentences is the first verb (V1), whose transitivity is crucial for establishing an initial preference for either an Active or a Passive interpretation as the sentence unfolds. Nonetheless, in line with the constraint-based approach to sentence processing and supported by previous research findings, a coherent structured interpretation of a sentence is determined by the combined constraints imposed by all words within that sentence. In our study, the transitivity of V1 alone is insufficient to fully explain the interpretative preference for the sentence structure. The overall sentence-level interpretation also depends on the thematic role preference of the subject noun – its likelihood of being an agent performing an action or a patient receiving the action.

This was evident in our findings, as shown in Author response image 1 above, where the V1 transitivity based on corpus or behavioural data did not fit to the neural data during the V1 epoch. In contrast, BERT structural measures [e.g., BERT parse depth vector (up to V1) and BERT V1 parse depth] offered contextualized representations that are presumed to integrate various lexical constraints present in each sentence. These BERT metrics exhibited significant model fits for the same neural data in the V1 epoch. Besides, a notable feature of BERT is its bi-directional attention mechanism, which allows for the dynamic updating of an earlier word’s representation as more of the sentence is heard, which is also changeling to achieve with corpus or behavioural metrics. For instance, the parse depth of the word “found” in the BERT parse depth vector for “The dog found…” differs from its parse depth in the vector for “The dog found in…”. This feature of BERT is particularly advantageous for investigating the dynamic nature of structured interpretation during speech comprehension, as it stimulates the continual updating of interpretation that occurs as a sentence unfolds (as shown by Figure 7 in the main text). We have elaborated on the rationale for employing BERT parse depth in this regard in the revised manuscript.

(3) The main data result figures comparing BERT and the EMEG brain data are hard to evaluate because only t-values are provided, and those, only for significant clusters. It would be helpful to see the full 600 ms time course of rho values, with error bars across subjects, to really be able to evaluate it visually. This is a summary statistic that is very far away from the input data

We appreciate this suggestion from R2. In the Appendix 1 of the revised manuscript, we have provided individual participants’ Spearman’s rho time courses for every model RDM tested in all the three epochs (see Appendix 1-figures 8-10 & 14-15). Note that RSA was conducted in the source-localized E/MEG, it is infeasible to plot the rho time course for each searchlight at one of the 8196 vertices on the cortical surface mesh. Instead, we plotted the rho time course of each ROI reported in the original manuscript. These plots complement the time-resolved heatmap of peak t-value in Figures 6-8 in the main text.

(4) Some details are omitted or not explained clearly. For example, how was BERT masked to give word-by-word predictions? In its default form, I believe that BERT takes in a set of words before and after the keyword that it is predicting. But I assume that here the model is not allowed to see linguistic information in the future.

In our analyses, we utilized the pre-trained version of BERT (Devlin et al. 2019) as released by Hugging Face (https://github.com/huggingface). It is noteworthy that BERT, as described in the original paper, was initially trained using the Cloze task, involving the prediction of masked words within an input. In our study, however, we neither retrained nor fine-tuned the pre-trained BERT model, nor did we employ it for word-by-word prediction tasks. We used BERT to derive the incremental representation of a sentence’s structure as it unfolded word-by-word.

Specifically, we sequentially input the text of each sentence into the BERT, akin to how a listener would receive the spoken words in a sentence (see Figure 3C in the main text). For each incremental input (such as “The dog found”), we extracted the hidden representations of each word from BERT. These representations were then transformed into their respective BERT parse depths using a structural probing model (which was trained using sentences with annotated dependency parse tress from the Penn Treebank Dataset). The resulting BERT parse depths were subsequently used to create model RDMs, which were then tested against neural data via RSA.

Crucially, in our approach, BERT was not exposed to any future linguistic information in the sentence. We never tested BERT parse depth of a word in an epoch where this word had not been heard by the listener. For example, the three-dimensional BERT parse depth vector for “The dog found” was tested in the V1 epoch corresponding to “found”, while the fourdimensional BERT parse depth vector for “The dog found in” was tested in the PP1 epoch of “in”.

How were the auditory stimuli recorded? Was it continuous speech or silences between each word? How was prosody controlled? Was it a natural speaker or a speech synthesiser?

Consistent with our previous studies (Kocagoncu et al. 2017; Klimovich-Gray et al. 2019; Lyu et al. 2019; Choi et al. 2021), all auditory stimuli in this study were recorded by a female native British English speaker, ensuring a neutral intonation throughout. We have incorporated this detail into the revised version of our manuscript for clarity.

It is difficult for me to fully assess the extent to which the authors achieved their aims, because I am missing important information about the setup of the experiment and the distribution of test statistics across subjects.

We are sorry for the previously omitted details regarding the experimental setup and the results of individual participants. As detailed in our responses above, we have now included the necessary information in the revised manuscript.

**Reviewer #3 (Public Review):**
Syntactic parsing is a highly dynamic process: When an incoming word is inconsistent with the presumed syntactic structure, the brain has to reanalyze the sentence and construct an alternative syntactic structure. Since syntactic parsing is a hidden process, it is challenging to describe the syntactic structure a listener internally constructs at each time moment. Here, the authors overcome this problem by (1) asking listeners to complete a sentence at some break point to probe the syntactic structure mentally constructed at the break point, and (2) using a DNN model to extract the most likely structure a listener may extract at a time moment. After obtaining incremental syntactic features using the DNN model, the authors analyze how these syntactic features are represented in the brain using MEG.

We extend our thanks to Reviewer #3 (referred to as R3 below) for recognizing the methods we used in this study.

Although the analyses are detailed, the current conclusion needs to be further specified. For example, in the abstract, it is concluded that "Our results reveal a detailed picture of the neurobiological processes involved in building structured interpretations through the integration across multifaceted constraints". The readers may remain puzzled after reading this conclusion.

Following R3’s suggestion, we have revised the abstract and refined our conclusions in the main text to explicitly highlight our principal findings. These include: (1) a shift from bihemispheric lateral frontal-temporal regions to left-lateralized regions in representing the current structured interpretation as a sentence unfolds, (2) a pattern of sequential activations in the left lateral temporal regions, updating the structured interpretation as syntactic ambiguity is resolved, and (3) the influence of lexical interpretative coherence activated in the right hemisphere over the resolved sentence structure represented in the left hemisphere.

Similarly, for the second part of the conclusion, i.e., "including an extensive set of bilateral brain regions beyond the classical fronto-temporal language system, which sheds light on the distributed nature of language processing in the brain." The more extensive cortical activation may be attributed to the spatial resolution of MEG, and it is quite well acknowledged that language processing is quite distributive in the brain.

We fully agree with R3 on the relatively low spatial resolution of MEG. Our emphasis was on the observed peak activations in specific regions outside the classical brain areas related to language processing, such as the precuneus in the default mode network, which are unlikely to be artifacts due to the spatial resolution of MEG. We have revised the relevant contents in the Abstract.

The authors should also discuss:(1) individual differences (whether the BERT representation is a good enough approximation of the mental representation of individual listeners).

To address the issue of individual differences which was also suggested by R2, we added individual participants’ model fits in ROIs with significant effects of BERT representations in Appendix 1 of the revised manuscript (see Appendix 1-figures 8-10 & 14-15).

(2) parallel parsing (I think the framework here should allow the brain to maintain parallel representations of different syntactic structures but the analysis does not consider parallel representations).

In the original manuscript, we did not discuss parallel parsing because the methods we used does not support a direct test for this hypothesis. In our analyses, we assessed the preference for one of two plausible syntactic structures (i.e., Active and Passive interpretations) based on the BERT parse vector of an incremental sentence input. This assessment was accomplished by calculating the mismatch between the BERT parse depth vector and the context-free dependency parse depth vector representing each of the two structures. However, we only observed one preferred interpretation in each epoch (see Figures 6D-6F) and did not find evidence supporting the maintenance of parallel representations of different syntactic structures in the brain. Nevertheless, in the revised manuscript, we have mentioned this possibility, which could be properly explored in future studies.

**Recommendations for the authors:**

**Reviewer #1 (Recommendations For The Authors):**
Consider fitting the behavioral data from the continuation pre-test to the brain data in order to illustrate the claimed advantage of using a computational model beyond more traditional methods.

Following R1’s suggestion, we conducted additional RSA using more behavioural and corpusbased metrics. We then directly compared the fits of these traditional metrics to brain data with those of BERT metrics in the same epoch to provide empirical evidence for the advantage of using a computational model like BERT to explain listeners’ neural data (see Appendix 1figures 11-13).

Clarify the use of "neural representations: For a clearer assessment of the results, please discuss your results (especially the fits with BERT parse depth) in terms of the potential effects of distinct sentence structure expectations on working memory demands and make clear where these can be disentangled from neural representations of an unfolding sentence's structure.

In the revised manuscript, we have noted the working memory demands associated with the online construction of a structured interpretation during incremental speech comprehension. As mentioned in our response to the relevant comment by R1 above, our experimental paradigm is not suitable for quantitatively assessing working memory demands since it is difficult to determine the exact number of open nodes for our stimuli with syntactic ambiguity before the disambiguating point (i.e., the main verb) is reached. Therefore, while we can speculate the potential contribution of varying working memory demands (which might correlate with BERT V1 parse depth) to RSA model fits, we think it is not possible to disentangle their effects from the neural representation of an unfolding sentence’s structure modelled by BERT parse depths in our current study.

Please add in methods a description of how the uniqueness point was determined.

In this study, we defined the uniqueness point of a word as the earliest point in time when this word can be fully recognized after removing all of its phonological competitors. To determine the uniqueness point for each word of interest, we first identified the phoneme by which this word can be uniquely recognized according to CELEX (Baayen et al. 1993). Then, we manually labelled the offset of this phoneme in the auditory file of the spoken sentence in which this word occurred. We have added relevant description of how the uniqueness point was determined in the Methods section of the revised manuscript.

I found the name "interpretative mismatch" very opaque. Maybe instead consider "preference".

We chose to use the term “interpretative mismatch” rather than “preference” based on the operational definition of this metric, which is the distance between a BERT parse depth vector and one of the two context-free parse depth vectors representing the two possible syntactic structures, so that a smaller distance value (or mismatch) signifies a stronger preference for the corresponding interpretation.

In the abstract, the authors describe the cognitive process under investigation as one of incremental combination subject to "multi-dimensional probabilistic constraint, including both linguistic and non-linguistic knowledge". The non-linguistic knowledge is later also referred to as "broad world knowledge". These terms lack specificity and across studies have been operationalized in distinct ways. In the current study, this "world knowledge" is operationalized as the likelihood of a subject noun being an agent or patient and the probability for a verb to be transitive, so here a more specific term may have been the "knowledge about statistical regularities in language".

In this study, we specifically define “non-linguistic world knowledge” as the likelihood of a subject noun assuming the role of an agent or patient, which relates to its thematic role preference. This type of knowledge is primarily non-linguistic in nature, as exemplified by comparing nouns like “king” and “desk”. Although it could be reflected by statistical regularities in language, thematic role preference hinges more on world knowledge, plausibility, or real-world statistics. In contrast, “linguistic knowledge” in our study refers to verb transitivity, which focuses on the grammatically correct usage of a verb and is tied to statistical regularities within language itself. In the revised manuscript, we have provided clearer operational definitions for these two concepts and have ensured consistent usage throughout the text.

Please spell out what exactly the "constraint-based hypothesis" is (even better, include an explicit description of the alternative hypothesis?).

The “constraint-based hypothesis”, as summarized in a review (McRae and Matsuki 2013), posits that various sources of information, referred to as “constraints”, are simultaneously considered by listeners during incremental speech comprehension. These constraints encompass syntax, semantics, knowledge of common events, contextual pragmatic biases, and other forms of information gathered from both intra-sentential and extra-sentential context.Notably, there is no delay in the utilization of these multifaceted constraints once they become available, neither is a fixed priority assigned to one type of constraint over another. Instead, a diverse set of constraints is immediately brought into play for comprehension as soon as they become available as the relevant spoken word is recognized.

An alternative hypothesis, proposed earlier, is the two-stage garden path model (Frazier and Rayner 1982; Frazier 1987). According to this model, there is an initial parsing stage that relies solely on syntax. This is followed by a second stage where all available information, including semantics and other knowledge, is used to assess the plausibility of the results obtained in the first-stage analysis and to conduct re-analysis if necessary (McRae and Matsuki 2013). In the Introduction of our revised manuscript, we have elaborated on the “constraint-based hypothesis” and mentioned this two-stage garden path model as its alternative.

Fig1 B&C: In order to make the data more interpretable, could you estimate how many possible grammatical structural configurations there are / how many different grammatical structures were offered in the pretest, and based on this what would be the "chance probability" of choosing a random structure or for example show how many responded with a punctuation vs alternative continuations?

In our analysis of the behavioural results, we categorized the continuations provided by participants in the pre-test at the offset of Verb1 (e.g., “The dog found/walked …”) into 6 categories, including DO (direct object), INTRANS (intransitive), PP (prepositional phrase), INF (infinitival complement), SC (sentential complement) and OTHER (gerund, phrasal verb, etc.).

**Author response table 1. sa3table1:** 

	HiTrans sentences	LoTrans sentences
DO	0.89±0.16	0.33±0.34
PP	0.04±0.08	0.39±0.29
INTRANS	0.02±0.05	0.21±0.24
INF	0.01±0.04	0.02±0.06
SC	SC 0.00±0.01	0.00±0.02
OTHER	0.04±0.08	0.06±0.12

Similarly, we categorized the continuations that followed the offset of the prepositional phrase (e.g., “The dog found/walked in the park …”) into 7 categories, including MV (main verb), END (i.e., full stop), PP (prepositional phrase), INF (infinitival complement), CONJ (conjunction), ADV (adverb) and OTHER (gerund, sentential complement, etc.).

**Author response table 2. sa3table2:** 

	HiTrans sentences	LoTrans sentences
MV	0.66±0.19	0.12±0.15
END	0.05±0.07	0.26±0.11
PP	0.04±0.05	0.19±0.12
INF	0.01±0.03	0.07±0.11
CONJ	0.03±0.05	0.12±0.10
ADV	0.01±0.03	0.08±0.10
OTHER	0.20±0.13	0.16±0.09

It is important to note that the results of these two pre-tests, including the types of continuations and their probabilities, exhibited considerable variability between and within each sentence type (see also Figures 2B and 2C).

Typo: "In addition, we found that BERT structural interpretations were also a correlation with the main verb probability" >> correlated instead of correlation.

We apologize for this typo. We have conducted a thorough proofreading to identify and correct any other typos present in the revised manuscript.

"In this regard, DLMs excel in a flexible combination of different types of features embedded in their rich internal representations". What are the "different types", spell out at least some examples for illustration.

We have rephrased this sentence to give a more detailed description.

Fig 2 caption: "Same color scheme as in (A)" >> should be 'as in (B)'?, and later A instead of B.

We are sorry for this typo. We have corrected it in the revised manuscript.

**Reviewer #2 (Recommendations For The Authors):**
My biggest recommendation is to make the paper clearer in two ways: (i) writing style, by hand-holding the reader through each section, and the motivation for each step, in both simple and technical language; (ii) schematic visuals, of the experimental design and the analysis. A schematic of the main experimental manipulation would be helpful, rather than just listing two example sentences. It would also be helpful to provide a schematic of the experimental setup and the analysis approach, so that people can refer to a visual aid in addition to the written explanation. For example, it is not immediately clear what is being correlated with what - I needed to go to the methods to understand that you are doing RSA across all of the trials. Make sure that all of the relevant details are explained, and that you motivate each decision.

We thank R2 for these suggestions. In the revised manuscript, we have enhanced the clarity of the main text by providing a more detailed explanation of the motivation behind each analysis and the interpretation of the corresponding results. Additionally, in response to R2’s recommendation, we have added a few figures, including the illustration of the experimental design (Figure 1) and methods (see Figure 3C and Figure 5).

Different visualisation of neural results - The main data result figures comparing BERT and the EMEG brain data are hard to evaluate because only t-values are provided, and those, are only for significant clusters. It would be helpful to see the full 600 ms time course of rho values, with error bars across subjects, to really be able to evaluate it visually.

In the original manuscript, we opted to present t-value time courses for the sake of simplicity in illustrating the fits of the 12 model RDMs tested in 3 epochs. Following R2’s suggestion, we have included the ROI model fit time courses of each model RDM for all individual participants, as well as the mean model fit time course with standard error in Appendix 1figures 8-10 & 14-15.

How are the authors dealing with prosody differences that disambiguate syntactic structures, that BERT does not have access to?

All spoken sentence stimuli were recorded by a female native British English speaker, ensuring a neutral intonation throughout. Therefore, prosody is unlikely to vary systematically between different sentence types or be utilized to disambiguate syntactic structures. Sample speech stimuli have been made available in the following repository: https://osf.io/7u8jp/.

A few writing errors: "was kept updated every time"

We are sorry for the typos. We have conducted proof-reading carefully to identify and correct typos throughout the revised manuscript.

Explain why the syntactic trees have "in park the" rather than "in the park"?

The dependency parse trees (e.g., Figure 3A) were generated according to the conventions of dependency parsing (de Marneffe et al. 2006).

Why are there mentions of the multiple demand network in the results? I'm not sure where this comes from.

The mention of the multiple demand network was made due to the significant RSA fits observed in the dorsal lateral prefrontal regions and the superior parietal regions, which are parts of the multiple demand network. This observation was particularly notable for the BERT parse depth vector in the main verb epoch when the potential syntactic ambiguity was being resolved. It is plausible that these effects observed are partly attributed to the varying working memory demands required to maintain the “opening nodes” in the different syntactic structures being considered by listeners at this point in the sentence.

**Reviewer #3 (Recommendations For The Authors):**
The study first asked human listeners to complete partial sentences, and incremental parsing of the partial sentences can be captured based on the completed sentences. This analysis is helpful and I wonder if the behavioral data here are enough to model the E/MEG responses. For example, if I understood it correctly, the parse depth up to V1 can be extracted based on the completed sentences and used for the E/MEG analysis.

The behavioural data alone do not suffice to model the E/MEG data. As we elucidated in our responses to R1, we employed three behavioural metrics derived from the continuation pretests. These metrics include the V1 transitivity and the PP probability, given the continuations after V1 (e.g., after “The dog found…”), as well as the MV probability, given the continuations after the prepositional phrase (e.g., after “The dog found in the park…”). These metrics aimed to capture participants’ prediction based on their structured interpretations at various positions in the sentence. However, none of these behavioural metrics yielded significant model fits to the listeners’ neural activity, which sharply contrasts with the substantial model fits of the BERT metrics in the same epochs. Besides, we also tried to model V1 parse depth as a weighted average based on participants’ continuations. As shown in Figure 3A, V1 parse depth is 0 in the active interpretation, 2 in the passive interpretation, while the parse depth of the determiner and the subject noun does not differ. However, this continuation-based V1 parse depth [i.e., 0 × Probability(active interpretation) + 2 × Probability(passive interpretation)] did not show significant model fits.

Related to this point, I wonder if the incremental parse extracted using BERT is consistent with the human results (i.e., parsing extracted based on the completed sentences) on a sentence-bysentence basis.

In fact, we did provide evidence showing the alignment between the incremental parse extracted using BERT and the human interpretation for the same partial sentence input (see Figure 4 in the main text and Appendix 1-figures 4-6).

Furthermore, in Fig 1d, is it possible to calculate how much variance of the 3 probabilities is explained by the 4 factors, e.g., using a linear model? If these factors can already explain most of the variance of human parsing, is it possible to just use these 4 factors to explain neural activity?

Following R3’s suggestion, we have conducted additional linear modelling analyses to compare the extent to which human behavioural data can be explained by corpus metrics and BERT metrics separately. Specifically, for each of the three probabilities obtained in the pretests (i.e., DO, PP, and MV), we constructed two linear models. One model utilized the four corpus-based metrics as regressors (i.e., SN agenthood, V1 transitivity, Passive index, and Active index), while the other model used BERT metrics as regressors (i.e., BERT parse depth of each word up to V1 from layer 13 for DO/PP probability and BERT parse depth of each word up to the end of PP from layer 14 for MV probability, consistent with the BERT layers reported in Figure 6).

As shown in the table below, corpus metrics demonstrate a more effective fit than BERT metrics for predicting the DO/PP probability. The likelihood of a DO/PP continuation is chiefly influenced by the lexical syntactic property of V1 (i.e., transitivity), and appears to rely less on contextual factors. Since V1 transitivity is explicitly included as one of the corpus metrics, it is thus expected to align more closely with the DO/PP probability compared to BERT metrics, primarily reflecting transitive versus intransitive verb usage.

**Author response table 3. sa3table3:** 

	Adjusted r^2^(corpora)	Adjusted r^2^(BERT)	F-statistic for model comparison
DO probability	0.28	0.06	F(1,115)=35.68, *P*=2.66 x 10^–8^
PP probability	0.18	–0.01	F(1,115)=28.62, *P*=4.54 x 10^–7^
MV probability	0.12	0.44	F(2,113)=33.21, *P*=4.51 x 10^–12^

Actually, BERT V1 parse depth was not correlated with V1 transitivity when the sentence only unfolds to V1 (see Appendix 1-figure 6). This lack of correlation may stem from the fact that the BERT probing model was designed to represent the structure of a (partially) unfolded sentence, rather than to generate a continuation or prediction. Moreover, V1 transitivity alone does not conclusively determine the Active or Passive interpretation by the end of V1. For instance, both transitive and intransitive continuations after V1 are compatible with an Active interpretation. Consequently, the initial preference for an Active interpretation (as depicted by the early effects before V1 was recognized in Figure 6D), might be predominantly driven by the animate subject noun (SN) at the beginning of the sentence, a word order cue in languages like English (Mahowald et al. 2023).

In contrast, when assessing the probability of a MV following the PP (e.g., after “The dog found in the park ...”), BERT metrics significantly outperformed corpus metrics in terms of fitting the MV probability. Although SN thematic role preference and V1 transitivity were designed to be the primary factors constraining the structured interpretation in this experiment, we could only obtain their context-independent estimates from corpora (i.e., considering all contexts). Additionally, despite Active/Passive index (a product of these two factors) are correlated with the MV probability, it may oversimplify the task of capturing the specific context of a given sentence. Furthermore, the PP following V1 is also expected to influence the structured interpretation. For instance, whether “in the park” is a more plausible scenario for people to find a dog or for a dog to find something. Thus, this finding suggests that the corpus-based metrics are not as effective as BERT in representing contextualized structured interpretations (for a longer sentence input), which might require the integration of constraints from every word in the input.

In summary, corpus-based metrics excel in explaining human language behaviour when it primarily relies on specific lexical properties. However, they significantly lag behind BERT metrics when more complex contextual factors come into play at the same time. Regarding their performance in fitting neural data, among the four corpus-based metrics, we only observed significant model fits for the Passive index in the MV epoch when the intended structure for a Passive interpretation was finally resolved, while the other three metrics did not exhibit significant model fits in any epoch. Note that subject noun thematic role preference did fit neural data in the PP and MV epochs (Figure 8A and 8B). In contrast, the incremental BERT parse depth vector exhibited significant model fits in all three epochs we tested (i.e., V1, PP1, and MV).

To summarize, I feel that I'm not sure if the structural information BERT extracts reflect the human parsing of the sentences, especially when the known influencing factors are removed.

Based on the results presented above and, in the manuscript, BERT metrics align closely with human structured interpretations in terms of both behavioural and neural data. Furthermore, they outperform corpus-based metrics when it comes to integrating multiple constraints within the context of a specific sentence as it unfolds.

Minor issues:Six types of sentences were presented. Three types were not analyzed, but the results for the UNA sentences are not reported either.

In this study, we only analysed two out of the six types of sentences, i.e., HiTrans and LoTrans sentences. The remaining four types of sentences were included to ensure a diverse range of sentence structures and avoid potential adaption the same syntactic structure.

Fig 1b, If I understood it correctly, each count is a sentence. Providing examples of the sentences may help. Listing the sentences with the corresponding probabilities in the supplementary materials can also help.

Yes, each count in Figure 2B (Figure 1B in the original manuscript) is a sentence. All sentence stimuli and results of pre-tests are available in the following repository https://osf.io/7u8jp/.

"trajectories of individual HiTrans and LoTrans sentences are considerably distributed and intertwined (Fig. 2C, upper), suggesting that BERT structural interpretations are sensitive to the idiosyncratic contents in each sentence." It may also mean the trajectories are noisy.

We agree with R3 that there might be unwanted noise underlying the distributed and intertwined BERT parse depth trajectories of individual sentences. Meanwhile, it is also important to note that the correlation between BERT parse depths and lexical constraints of different words at the same position across sentences is statistically supported.

References

Baayen RH, Piepenbrock R, van H R. 1993. The {CELEX} lexical data base on {CD-ROM}. Baroni M, Dinu G, Kruszewski G. 2014. Don't count, predict! A systematic comparison of contextcounting vs. context-predicting semantic vectors. Proceedings of the 52nd Annual Meeting of the Association for Computational Linguistics, Vol 1.238-247.

Caucheteux C, King JR. 2022. Brains and algorithms partially converge in natural language processing. Communications Biology. 5:134.

Choi HS, Marslen-Wilson WD, Lyu B, Randall B, Tyler LK. 2021. Decoding the Real-Time Neurobiological Properties of Incremental Semantic Interpretation. Cereb Cortex. 31:233-247.

de Marneffe M-C, MacCartney B, Manning CD editors. Generating typed dependency parses from phrase structure parses, Proceedings of the 5th International Conference on Language Resources andEvaluation; 2006 May 22-28, 2006; Genoa, Italy:European Language Resources Association. 449-454 p.

Devlin J, Chang M-W, Lee K, Toutanova K editors. BERT: Pre-training of Deep Bidirectional Transformers for Language Understanding, Proceedings of the 2019 Conference of the North American Chapter of the Association for Computational Linguistics: Human Language Technologies; 2019 June 2-7, 2019; Minneapolis, MN, USA:Association for Computational Linguistics. 4171-4186 p.

Frazier L. 1987. Syntactic processing: evidence from Dutch. Natural Language & Linguistic Theory. 5:519-559.

Frazier L, Rayner K. 1982. Making and correcting errors during sentence comprehension: Eye movements in the analysis of structurally ambiguous sentences. Cognitive Psychology. 14:178-210.

Klimovich-Gray A, Tyler LK, Randall B, Kocagoncu E, Devereux B, Marslen-Wilson WD. 2019. Balancing Prediction and Sensory Input in Speech Comprehension: The Spatiotemporal Dynamics of Word Recognition in Context. Journal of Neuroscience. 39:519-527.

Kocagoncu E, Clarke A, Devereux BJ, Tyler LK. 2017. Decoding the cortical dynamics of soundmeaning mapping. Journal of Neuroscience. 37:1312-1319.

Lyu B, Choi HS, Marslen-Wilson WD, Clarke A, Randall B, Tyler LK. 2019. Neural dynamics of semantic composition. Proceedings of the National Academy of Sciences of the United States of America. 116:21318-21327.

Mahowald K, Diachek E, Gibson E, Fedorenko E, Futrell R. 2023. Grammatical cues to subjecthood are redundant in a majority of simple clauses across languages. Cognition. 241:105543.

McRae K, Matsuki K. 2013. Constraint-based models of sentence processing. Sentence processing. 519:51-77.

Schrimpf M, Blank IA, Tuckute G, Kauf C, Hosseini EA, Kanwisher N, Tenenbaum JB, Fedorenko E. 2021. The neural architecture of language: Integrative modeling converges on predictive processing. Proceedings of the National Academy of Sciences of the United States of America. 118:e2105646118.